# deCIFer: Crystal Structure Prediction from Powder Diffraction Data using Autoregressive Language Models

**Frederik Lizak Johansen**                                                    *frjo@di.ku.dk*
*Department of Computer Science, University of Copenhagen, Denmark*

**Ulrik Friis-Jensen**                                                          *ufj@chem.ku.dk*
*Department of Chemistry & Nano-Science Center, University of Copenhagen, Denmark*

**Erik Bjørnager Dam**                                                       *erikdam@di.ku.dk*
*Department of Computer Science, University of Copenhagen, Denmark*

**Kirsten Marie Ørnsbjerg Jensen**                                        *kirsten@chem.ku.dk*
*Department of Chemistry & Nano-Science Center, University of Copenhagen, Denmark*

**Rocío Mercado**                                                            *rocio@ailab.bio*
*Department of Computer Science & Engineering, Chalmers University of Technology, Sweden*

**Raghavendra Selvan**                                                       *raghav@di.ku.dk*
*Department of Computer Science, University of Copenhagen, Denmark*

**Reviewed on OpenReview:** *https://openreview.net/forum?id=LftFQ35l47*

## Abstract

Novel materials drive advancements in fields ranging from energy storage to electronics, with crystal structure characterization forming a crucial yet challenging step in materials discovery. In this work, we introduce *deCIFer*, an autoregressive language model designed for powder X-ray diffraction (PXRD)-conditioned crystal structure prediction (PXRD-CSP). Unlike traditional CSP methods that rely primarily on composition or symmetry constraints, deCIFer explicitly incorporates PXRD data, directly generating crystal structures in the widely adopted Crystallographic Information File (CIF) format. The model is trained on nearly 2.3 million crystal structures, with PXRD conditioning augmented by basic forms of synthetic experimental artifacts, specifically Gaussian noise and instrumental peak broadening, to reflect fundamental real-world conditions. Validated across diverse synthetic datasets representative of challenging inorganic materials, deCIFer achieves a 94% structural match rate. The evaluation is based on metrics such as the residual weighted profile ($R_{wp}$) and structural match rate (MR), chosen explicitly for their practical relevance in this inherently underdetermined problem. deCIFer establishes a robust baseline for future expansion toward more complex experimental scenarios, bridging the gap between computational predictions and experimental crystal structure determination.

## 1 Introduction

Characterizing the atomic structure of functional materials is essential for enabling progress in energy storage, electronics, and other emerging technologies. Powder X-ray diffraction (PXRD) is a widely employed experimental technique for this purpose: it measures a one-dimensional intensity profile whose peaks reflect the periodic arrangement of atoms in a crystalline solid. This makes structure determination an inverse problem: a three-dimensional periodic structure is mapped to a one-dimensional signal by a (largely) known forward process, and the inverse is often underdetermined, so multiple structures can match similar PXRD

profiles. In experimental crystallography, structure solution from PXRD is typically carried out through a multi-stage workflow: preprocessing and peak finding, proposing an initial structural hypothesis, and then *Rietveld refinement*, which is a nonlinear least-squares fit of a simulated PXRD profile to the measured profile. The initial hypothesis is often obtained by identifying representative candidate structures from crystallographic databases, and using these as starting points for refinement. A key bottleneck is obtaining a good initial candidate, especially when peaks overlap, the pattern is noisy, or the correct structure type is not well represented in available databases.

Crystal structure prediction (CSP) refers to the computational task of inferring these same structures given a set of constraints or observations. In traditional settings, these constraints are limited to high-level descriptors such as chemical composition or symmetry. While recent work has explored integrating experimental data into generative models, direct PXRD-conditioned CSP remains underexplored. Public structure databases predominantly archive refined structures, but do not routinely provide standardized, model-ready paired PXRD patterns, since obtained data scans depend on instrument settings, sample preparation, and preprocessing choices. As a result, large aligned (structure, PXRD) corpora are uncommon, which has limited end-to-end PXRD conditioning in generative models.

In this work, we present *deCIFer*, an autoregressive transformer-based model designed explicitly for PXRD-conditioned crystal structure prediction (PXRD-CSP) (Figure 1a). To study PXRD conditioning despite the limited availability of paired experimental data, we construct a controlled paired setting: for each reference crystal structure we generate its corresponding PXRD profile using a defined forward simulator, and we use these paired examples for large-scale training and reproducible evaluation. This provides a controlled setting for large-scale training and evaluation, enabling our study of PXRD-conditioned generation. deCIFer can then be extended as richer experimental structure-PXRD datasets become available.

Given a target PXRD profile, deCIFer generates a complete *crystallographic structure description*. We represent this description using the *Crystallographic Information File (CIF)* format, a standard structured text-representation used to exchange crystal structures and to interface with common crystallography tools, including refinement pipelines and structure databases. At the same time, this motivates a decoder-only autoregressive design: CIFs are variable-length structured text with strict syntax, and autoregressive decoding both supports fixing known CIF descriptors (such as chemical composition) during generation and enables sampling multiple PXRD-consistent hypotheses that can be ranked by PXRD-consistency. This enables controlled comparisons (PXRD-only vs. PXRD+descriptors). deCIFer is intended to complement traditional workflows as a hypothesis generator, proposing PXRD-consistent candidate structures which can then be filtered, ranked by PXRD-consistency, and refined using standard tools.

To establish a practical baseline, we incorporate basic forms of synthetic experimental artifacts into our training and validation datasets, specifically Gaussian noise and instrumental peak broadening. These basic synthetic artefacts represent fundamental yet simplified aspects of real-world PXRD variability, chosen intentionally to provide a controlled, reproducible starting point for evaluation and for future extensions of the method. They act as a minimal simulation of measurement noise and resolution limits, analogous to data augmentation strategies in vision or speech domains. They do not capture more complex effects such as peak asymmetry, background drift, or preferred orientation, which we leave for future studies.

The performance of deCIFer is demonstrated on diverse PXRD patterns from large-scale datasets representative of challenging inorganic materials. Our evaluations confirm the robustness of the approach and its ability to produce syntactically correct and structurally meaningful CIFs that accurately reproduce target diffraction patterns. With this, deCIFer serves as a foundational step toward bridging the gap between computational CSP and experimental crystal structure determination workflows.

Our key contributions in this work are:

1. Integration of PXRD-based experimental conditioning into an autoregressive transformer model, enabling direct CSP in CIF format; a capability not demonstrated previously in transformer-based generative models.
2. Implementation of an effective conditioning mechanism to handle variable-length CIF sequences in autoregressive modelling.

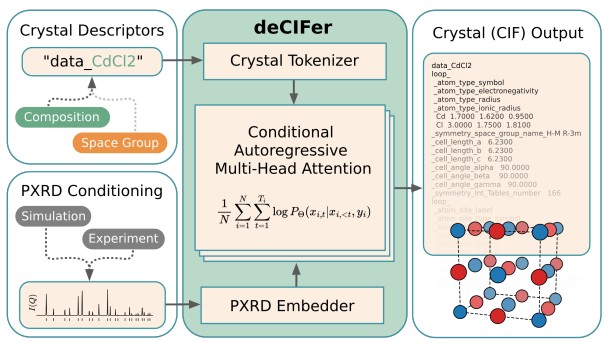

(a) Overview of the deCIFer model.

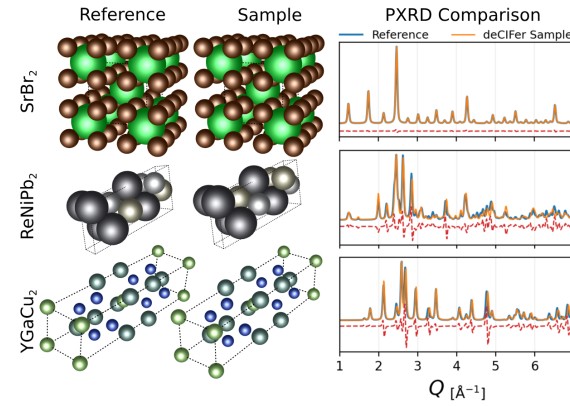

(b) Example generations using deCIFer.

Figure 1: (a) Overview of the deCIFer model, which performs autoregressive crystal structure prediction (CSP) from PXRD data, optionally guided by standard crystallographic descriptors (e.g., composition and space group) encoded as CIF fields, which can be provided to constrain generation or omitted for exploratory sampling. PXRD embeddings are prepended to the CIF token sequence, enabling the generation of structurally consistent CIFs directly from diffraction data. (b) Three examples from the NOMA test set showing deCIFer generations, each illustrating a reference structure, the generated structure and their corresponding PXRD profiles.

3. Simulation of fundamental PXRD experimental artefact, Gaussian noise and peak broadening, as a practical baseline for real-world scenarios.

4. Comprehensive evaluation on two large-scale datasets: NOMA[1] and CHILI-100K (Friis-Jensen et al., 2024), including comparison to state-of-the-art CSP models and analysis of sampling consistency under varying conditions.

These contributions aim to establish a reproducible foundation for integrating computational CSP with experimental workflows.

## 2  Background and Related Work

CSP has traditionally relied on high-level descriptors, such as chemical composition or symmetry constraints, to guide predictions. Recently, approaches termed *data-informed* CSP have begun integrating explicit experimental data (particularly diffraction data) into their generative processes (Kjær et al., 2023; Guo et al., 2025; Riesel et al., 2024; Lai et al., 2025), marking a significant shift away from purely descriptor-driven CSP. Among the many experimental modalities that could inform such models, PXRD is especially relevant because it directly encodes crystallographic information in a widely accessible format. This makes it a natural focus for the present work.

**PXRD:** Powder X-ray diffraction (PXRD) is among the most widely accessible and routinely employed structural characterization techniques in solid-state chemistry. Modern benchtop diffractometers, available in most research and industrial laboratories, can produce high-quality diffraction patterns in minutes. PXRD patterns consist of diffraction peaks whose positions and intensities directly encode essential information about a material's crystal structure; specifically atomic arrangements and symmetry. The forward simulation of PXRD patterns from known crystal structures in the standard Crystallographic Information File (CIF) format is well-established through scattering theory (West, 2014), facilitating realistic computational modelling.

Quantitative PXRD analysis typically involves structural refinement methods, such as Rietveld refinement (Young, 1995), wherein parameters of a structural model are iteratively adjusted against experimental

---

[1]NOMA stands for **N**OMAD (Draxl & Scheffler, 2019), **O**QMD (Kirklin et al., 2015), & **M**P (Jain et al., 2013b) **A**ggregation.

data. Such refinements depend heavily on the accuracy of an initial structural model, whose identification (often termed *fingerprinting*) can be particularly challenging. Fingerprinting typically requires extensive chemical intuition and exhaustive database searches. Nevertheless, structural model identification often remains ambiguous, significantly hindering materials discovery and optimization.

**CSP with LLMs:** Large language models (LLMs) based on transformer architectures (Vaswani et al., 2017) have recently been leveraged for automation tasks in chemistry, including synthesis planning (Hocky & White, 2022; Szymanski et al., 2023; M. Bran et al., 2024), chemical data extraction (Gupta et al., 2022; Dagdelen et al., 2024; Polak & Morgan, 2024; Schilling-Wilhelmi et al., 2025), and property prediction (Zhang et al., 2024; Rubungo et al., 2024; Jablonka et al., 2024). Despite their growing popularity, these LLMs have yet to become widely utilized in materials design workflows. Recent work has started adapting these models specifically for crystal structure prediction (CSP). For instance, Gruver et al. (2024) fine-tuned Llama-2 models (Touvron et al., 2023) on text-based representations of atomistic data, enabling tasks such as the unconditional generation of stable crystalline materials. Similarly, Mohanty et al. (2024) fine-tuned LLaMA-3.1-8B (Dubey et al., 2024) using QLoRA (Dettmers et al., 2023) to efficiently generate CIFs conditioned on compositional and symmetry constraints. Another recent approach, CrystaLLM (Antunes et al., 2024), utilizes extensive pre-training on millions of inorganic crystal structures and likewise targets CIF generation using only high-level descriptors like composition and symmetry, without explicitly incorporating experimental constraints. While these methods represent significant advancements in generative CSP, they remain disconnected from direct experimental observations, which are often critical for accurate and practical structure determination.

**CSP with diffusion models:** In parallel to transformer-based LLM approaches, diffusion- and flow-based generative models have emerged as complementary methods for CSP (Jiao et al., 2023; Miller et al., 2024; Zeni et al., 2025; Xie et al., 2022). These frameworks typically utilize compositional constraints or partial structural information to guide structure generation and have shown promise in reliably predicting stable crystalline configurations. However, like many transformer-based CSP methods, diffusion-based models predominantly rely on purely computational constraints. The recent framework MatterGen (Zeni et al., 2025) has made notable strides by enabling generative modeling conditioned on a variety of property-based constraints, improving predictions for structures likely to be synthesizable. Nevertheless, direct conditioning on experimental data, such as PXRD patterns, remains underexplored in diffusion-based CSP, underscoring the necessity for methods that bridge computational generation with explicit experimental data conditioning.

## 3 Methods

Consider a crystal structure represented in the CIF format, tokenized into a sequence of length $T_i$: $\mathbf{x}^i = (x_1^i, x_2^i, \ldots, x_{T_i}^i)$ (see Appendix A.3 for details). The corresponding PXRD pattern, denoted by $\mathbf{y}^i$, is a continuous-valued vector encoding the intensity profile of the diffraction pattern. Our dataset thus comprises pairs of CIF sequences and their corresponding PXRD patterns: $\mathcal{D} = [(\mathbf{x}^i, \mathbf{y}^i)]_{i=1}^N$. Our objective is to minimize the negative conditional log-likelihood over this training data:

$$\mathcal{L}(\mathbf{X}|\mathbf{Y};\mathbf{\Theta}) = \frac{1}{N}\sum_{i=1}^{N}\left(-\sum_{t=1}^{T_i}\log P_{\mathbf{\Theta}}(x_t^i|x_{<t}^i, \mathbf{y}^i)\right). \tag{1}$$

This is accomplished through the transformer-based conditional autoregressive model $f_{\mathbf{\Theta}}(\cdot)$, termed *deCIFer*. Given PXRD data, deCIFer generates crystal structures in the CIF format autoregressively.

**PXRD Conditioning:** PXRD data explicitly encodes structural fingerprints of crystal structures. We leverage PXRD as a direct conditioning input to guide our CSP.

Using standard crystallographic procedures, we generate discrete diffraction peaks from CIF structures, represented as the set $\mathcal{P} = \{(q_k, i_k)\}_{k=1}^n$, via `pymatgen` (Ong et al., 2013). These peaks are transformed into continuous PXRD patterns, $\mathbf{y}$, under synthetic experimental conditions. Formally, let $\mathcal{T}$ represent a set of transformations applied to $\mathcal{P}$.

To establish a robust baseline model, our transformations reflect simplified but fundamental experimental artifacts: each transformation $\tau \sim \mathcal{T}$ consists of (1) peak broadening characterized by a full-width-at-half-maximum (FWHM) sampled uniformly from 0.001 to 0.100, and (2) additive Gaussian noise with variance $\sigma_{\text{noise}}^2$ uniformly sampled from 0.001 to 0.050. A new $\tau$ is sampled for each training sample on every epoch, ensuring diverse exposure to synthetic experimental variability. For evaluation purposes, we define a fixed transformation, $\tau_{\text{fixed}}$, with specific parameters governed by the experiments to systematically assess robustness, and a *clean* transformation $\tau_0$ (FWHM $= 0.05$, $\sigma_{\text{noise}}^2 = 0$) to assess similarity in context of PXRD. Examples from $\mathcal{T}$ on a PXRD are shown in Figure A3 (in Appendix).

**Conditioning Model:** PXRD patterns are embedded into a learned vector space via a multilayer perceptron (MLP) $f_{\mathbf{\Phi}}(\mathbf{y})$, parameterized by $\mathbf{\Phi}$. The resulting embedding vector $\mathbf{e} = f_{\mathbf{\Phi}}(\mathbf{y}) \in \mathbb{R}^D$ is prepended to the tokenized CIF sequence, providing a direct conditioning mechanism. Joint optimization of the embedding network $f_{\mathbf{\Phi}}$ and the transformer model $f_{\mathbf{\Theta}}$ results in our final objective: $\mathcal{L}(\mathbf{X}|\mathbf{Y}; \mathbf{\Theta}, \mathbf{\Phi})$.

**Sequence Packing and Isolation:** To handle variable-length CIF sequences efficiently during training, we employ sequence packing, inspired by recent methods in NLP (Kosec et al., 2021). Tokenized CIF sequences, each of length $T_i$, are concatenated into fixed-length segments of context size $C = 3076$, chosen to optimize GPU usage and throughput. Formally, a packed sequence is represented as $\mathbf{S} = [\mathbf{e}^1, \mathbf{t}_1^1, \ldots, \mathbf{t}_{T_1}^1, \mathbf{e}^2, \mathbf{t}_1^2, \ldots, \mathbf{t}_k^n]$, where each $\mathbf{e}^i$ is the $D$-dimensional conditioning embedding, and $\mathbf{t}_j^i$ are input embeddings. Long CIFs exceeding $C$ tokens are split between sequences inside batches, but occur infrequently ($\approx 0.04\%$ of sequences; see Figure A4 in the Appendix). To reduce adverse effects from splits, data shuffling is performed each epoch.

Isolation between CIFs within a packed sequence is ensured by an attention mask $\mathbf{M}$, where $M_{kl} = 1$ if tokens $k$ and $l$ originate from the same CIF, and 0 otherwise, resulting in block-wise diagonal attention structures (shown in Figure A2 in the Appendix). Additionally, positional encodings are reset at each CIF boundary, preventing leakage of positional information across CIF sequences.

# 4 Dataset and Experiments

**Dataset:** We utilize two large-scale open-source datasets that serve as the foundation for this study. The first, **NOMA**, is a synthetic dataset comprising crystal structures aggregated in CrystaLLM (Antunes et al., 2024), sourced from the Materials Project (April 2022) (Jain et al., 2013b), OQMD (v. 1.5, October 2023) (Kirklin et al., 2015), and NOMAD (April 2023) (Draxl & Scheffler, 2019). The second, **CHILI-100K** (Friis-Jensen et al., 2024), contains experimentally determined structures curated from a filtered subset of the Crystallography Open Database (COD) (Gražulis et al., 2009). NOMA is used for both training and testing, while CHILI-100K is used *exclusively for testing*. Both datasets are open-source and available for download.[2]

These datasets are intentionally chosen to approximate key aspects of real-world PXRD structure determination under controlled settings. Although synthetic in nature, they represent a practical and reproducible foundation for benchmarking models under basic structural and experimental variability. We note that these datasets are not intended to span the full spectrum of real experimental complexity. Rather, they are used here to establish a robust baseline for future studies that incorporate richer experimental variation and direct measurements.

**Preprocessing:** We follow the standard CrystaLLM preprocessing pipeline and apply additional steps to ensure consistency between NOMA and CHILI-100K. For NOMA, we select the lowest-volume structure per composition, filter duplicates, and retain only fully occupied CIFs with standardized formatting. The resulting dataset comprises approximately 2.3 million structures containing between 1–10 elements, up to atomic number 94 (excluding unstable or radioactive elements). Floating point values are rounded to four decimal places. For CHILI-100K, we retain $\approx 8{,}200$ experimentally derived CIFs with up to 8 elements, including atoms up to atomic number 85. Detailed statistics including space group distributions, token lengths, and composition diversity are provided in Appendix Figures A4 and A5.

---

[2]NOMA: `github.com/lantunes/CrystaLLM` (CC-BY 4.0 licence), CHILI-100K: `github.com/UlrikFriisJensen/CHILI` (Apache 2.0 licence).

Due to the known overrepresentation of high-symmetry structures in synthetic databases (Davariashtiyani et al., 2024; Zhang et al., 2023), we apply stratified sampling during the NOMA train/validation/test split based on space group labels. This mitigates structural distributional biases and improves evaluation robustness. For further details, see Section A.9.

**Tokenization:** All CIFs are tokenized using a 373-token vocabulary, including space group and element symbols, CIF tags, numerics, punctuation, and conditioning tokens. See Sections A.2 and A.3 in the Appendix for full preprocessing details.

**Model Hyperparameters:** deCIFer consists of two components: the PXRD encoder $f_{\Phi}$, a 2-layer MLP that maps a 1000-dimensional PXRD profile into a 512-dimensional embedding; and the structure generator $f_{\Theta}$, an 8-layer decoder-only transformer (Vaswani et al., 2017) with 8 attention heads per layer. The token dimension is set to 512 for both components. The model is trained using AdamW (Loshchilov & Hutter, 2017) with a batch size of 32 and a context length of 3076. Learning rate is linearly warmed up over the first 100 steps, followed by a cosine decay over 50,000 steps. Gradient accumulation (40 steps) and mixed-precision training are used on a single NVIDIA A100 GPU. These optimization and scheduling hyperparameters follow CrystaLLM (Antunes et al., 2024), which we adopt as a previously validated default for autoregressive CIF generation.

The total parameter count is 27.72M: $f_{\Phi}$ has $\approx 0.78$M and $f_{\Theta}$ has $\approx 26.94$M. All components are implemented in PyTorch (Paszke et al., 2019). Full architectural and training details are provided in Section A.9.

**Evaluation:** Figure 2 outlines the evaluation procedure. A reference CIF from the test set is first converted into a discrete set of diffraction peaks, $\mathcal{P} = \{(q_k, i_k)\}_{k=1}^{n}$, which are then transformed into a continuous PXRD pattern $\mathbf{y} = \tau_{\text{fixed}}(\mathcal{P})$, where $\tau_{\text{fixed}}$ simulates a predefined experimental setting. This PXRD signal, together with any known descriptors (e.g., composition or space group) when available, is passed to deCIFer to generate a new CIF structure, CIF*. All evaluations are based on a clean transformation $\tau_0$ with FWHM = 0.05 and no added noise.

We evaluate the generated structures using three complementary metrics:

(1) *Residual Weighted Profile* ($R_{\text{wp}}$) quantifies the difference between the reference and generated PXRD profiles. It is computed as $R_{\text{wp}} = \sqrt{\frac{\sum_i (y_i - y_i^*)^2}{\sum_i y_i^2}}$, where all weights $w_i = 1$, following convention.

(2) *Match Rate (MR)* measures geometric similarity using `StructureMatcher` (Ong et al., 2013). Two CIFs are considered a match if their lattice, atomic positions, and symmetry match within tolerance thresholds. MR is the fraction of matches over the total number of test structures. Full details are given in Appendix Section A.7.

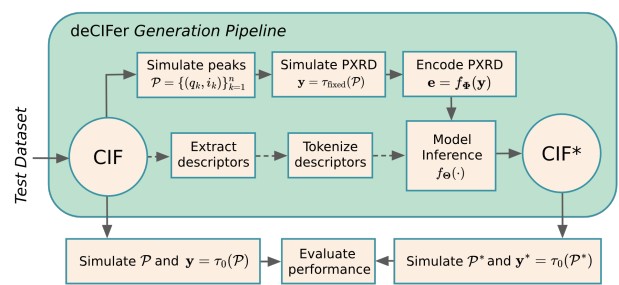

Figure 2: Evaluation pipeline: A test set CIF generates a PXRD profile, tokenized for deCIFer to produce a new CIF, compared to the reference using a clean transformation.

(3) *Validity (Val.)* assesses whether the generated CIF is internally consistent. A structure is considered valid only if it passes all four checks: formula balance, site multiplicity, realistic bond lengths, and symmetry agreement. See Appendix Section A.6 for criteria.

These metrics were chosen for their practical relevance in evaluating structure predictions from PXRD data. While each carries limitations, they jointly capture fidelity to both diffraction signals and crystallographic constraints.

**Experiments:** We designed four controlled experiments to assess deCIFer's ability to perform PXRD-conditioned CSP across a range of settings:

(1) *Comparison with State-of-the-Art:* We benchmark deCIFer against three recent CSP models that rely solely on composition or symmetry constraints, evaluating the benefit of PXRD-based conditioning. We report match rate (MR) under the standard composition-only protocol to enable a like-for-like comparison.

(2) *Ablation of PXRD Conditioning:* We compare deCIFer with an unconditioned variant, U-deCIFer, to isolate the contribution of PXRD data to the quality of generated structures. Here, diffraction-profile agreement ($R_{wp}$) is the primary measure of the contribution of conditioning, with MR and validity reported as complementary checks.

(3) *Robustness to Perturbations:* We apply deCIFer to PXRD data with varying levels of synthetic noise and peak broadening, simulating more challenging, experimentally relevant condition.

(4) *Generalization to CHILI-100K:* We evaluate deCIFer (trained on NOMA) on the CHILI-100K dataset to assess generalization to more chemically and structurally diverse experimental data.

In all experiments, we generate one structure per PXRD input. We also vary which information is provided at inference time. We use three descriptor settings: "none" (no fixed descriptors), "comp." (composition fixed), and "comp.+s.g." (composition and space group fixed), implemented by fixing the corresponding CIF fields during generation (see Section A.3). In the special case U-deCIFer + "none", the model receives no fixed descriptors and is initialized only from the CIF start token, and therefore samples unconditionally from its learned prior over CIFs. For this case, a near-zero match rate is to be expected, and we interpret it primarily via validity and $R_{wp}$.

## 5 Results

**Baseline Comparisons with State-of-the-art:** We compare deCIFer with three state-of-the-art CSP models that generates structures solely from the composition descriptor: CDVAE (Xie et al., 2022), DiffCSP (Jiao et al., 2023), and CrystaLLM (Antunes et al., 2024).

To compare these models within a shared evaluation setup, we follow the protocol used in prior work (Antunes et al., 2024): for each test composition, a single structure is generated and evaluated using match rate and atomic root-mean-square error (RMSE). Table 1 shows the results. For deCIFer and U-deCIFer, we report mean $\pm$ 1 standard deviation over three independent training runs (different random seeds); other methods are shown as reported in prior work.

The baseline models are provided only the composition. Nevertheless, they achieve high match rates. This can happen when a composition is strongly associated with one common structure type in the training data, so returning the most frequent structure often matches the test reference. However, the same composition can also have multiple different crystal structures (*polymorphs*) with the same elements but different atomic arrangements. In those cases, composition alone is not enough to select the structure that would be observed in a particular experiment.

deCIFer, by contrast, can only partially rely on compositional priors and is explicitly constrained by the PXRD signal, which defines a more structurally constrained and underdetermined prediction task. As shown in Table 1, deCIFer achieves substantially higher match rates on Perov-5 and Carbon-24, where the PXRD patterns offer strong structural signals. On MP-20 and MPTS-52, however, the match rate drops significantly. This performance drop may stem from the fact that the PXRD signal, when weak or ambiguous, does not sufficiently narrow the space of plausible structures; or worse, it may mislead the model away from the correct solution. In such cases, deCIFer appears unable to resolve the true structure from the PXRD input, suggesting that conditioning does not always help, and in some cases might even interfere with the model's prior-based predictions.

Still, this failure is instructive. It highlights the genuine difficulty of structure determination in the presence of polymorphism and non-discriminative PXRD data. This is a problem that naturally includes failure modes, ambiguity, and uncertainty, all of which are essential elements in real-world structure determination. Unlike traditional CSP models, which confidently return the most likely structure according to their priors, deCIFer conditions generation on an observed measurement.

Table 1: Performance comparison on four public CSP benchmarks: Perov-5 (Castelli et al., 2012a;b), Carbon-24 (Pickard, 2020), MP-20 (Jain et al., 2013a), and MPTS-52 (Baird, 2023). Following the single-sample evaluation protocol used in prior work (Antunes et al., 2024), one structure is generated per test composition. We report the match rate (%) based on structural equivalence under `StructureMatcher` and the atomic root-mean-square error (RMSE in Å) after alignment. For deCIFer and U-deCIFer we report mean $\pm$ 1 standard deviation over three independent seeds; other baselines are single-run values taken from the original references.

| | Perov-5 | | Carbon-24 | | MP-20 | | MPTS-52 | |
|---|---|---|---|---|---|---|---|---|
| Model | Match (%)↑ | RMSE↓ | Match (%)↑ | RMSE↓ | Match (%)↑ | RMSE↓ | Match (%)↑ | RMSE↓ |
| CDVAE | 45.31 | 0.1138 | 17.09 | 0.2969 | 33.90 | 0.1045 | 5.34 | 0.2106 |
| DiffCSP | 52.02 | 0.0760 | 17.54 | 0.2759 | 51.49 | 0.0631 | 12.19 | 0.1786 |
| CrystaLLM-small | 47.95 | 0.0966 | 21.13 | 0.1687 | 55.85 | 0.0437 | 17.47 | 0.1113 |
| CrystaLLM-large | 46.10 | 0.0953 | 20.25 | 0.1761 | 58.70 | 0.0408 | 19.21 | 0.1110 |
| U-deCIFer | 51.55±0.99 | 0.1113±0.0064 | 17.60±0.27 | 0.1532±0.0006 | 44.78±0.21 | 0.0777±0.0007 | 11.70±0.10 | 0.1547±0.0016 |
| **deCIFer** | 85.59±0.38 | 0.0480±0.0015 | 37.96±0.62 | 0.2028±0.0045 | 44.65±1.40 | 0.0709±0.0033 | 11.60±0.13 | 0.1444±0.0070 |

Overall, Table 1 shows that run-to-run variability in match rate is narrow across random seeds. However, as we will see later, repeating the inference step multiple times for the same PXRD input reveals a different kind of variability: when the PXRD conditioning signal is inconclusive, samples drawn from that single input do not collapse to one answer, but instead exhibit increased sample-to-sample diversity. This effect is most pronounced in more ambiguous regimes, such as lower-symmetry systems, where the PXRD profile is less discriminative and multiple structural solutions are plausible.

**Importance of PXRD Conditioning:** Having established the challenges of PXRD-CSP in ambiguous cases, we now examine the effect of PXRD conditioning within a controlled synthetic setting. Specifically, we compare deCIFer with its unconditioned variant (U-deCIFer) on the NOMA test set, using simulated PXRD profiles with controlled noise and broadening. This ablation isolates the impact of PXRD input by evaluating both models under identical architectural conditions, varying only in the inclusion of structural conditioning.

We consider three descriptor settings: (i) no additional input ("none"), (ii) composition only ("comp."), and (iii) composition plus space group ("comp. + s.g."). Figure 3 shows that deCIFer consistently outperforms U-deCIFer across all settings in terms of $R_{\mathrm{wp}}$, indicating a closer match between the generated and reference PXRD profiles. While U-deCIFer benefits from access to composition and symmetry descriptors, it never reaches the fidelity achieved by PXRD-conditioned generation.

The improvement is particularly pronounced in the absence of any crystal descriptors, where deCIFer significantly outperforms U-deCIFer. This demonstrates that PXRD alone provides a strong structural signal, whereas unconditioned generation collapses without access to priors. The gains persist when descriptors are included, with PXRD further narrowing the solution space toward structures that reproduce the target pattern.

Figure 4 further supports this, showing that performance is best for common, high-symmetry crystal systems, while rare or low-symmetry systems remain more difficult. The three test examples illustrate the spectrum of outcomes: from precise structural matches to clear mismatches, reflecting the varying information content in the PXRD input.

**Robustness to Perturbations in PXRD Conditioning:** To establish a reproducible baseline for PXRD-conditioned generation, we probe robustness under two simple, parameterized perturbation families applied to simulated PXRD patterns: additive Gaussian noise and instrumental peak broadening. These are the same perturbation families used during training, and we use them here as interpretable proxies for degraded signal-to-noise ratio and finite resolution. We evaluate performance across fixed parameter settings that are explicitly defined relative to the training perturbation ranges: in-distribution (ID) settings use noise and broadening values that lie within the ranges sampled during training, whereas out-of-distribution (OOD) settings use values outside those training ranges to test extrapolation to more severe corruption. We do this within the bounds of the assumed forward simulator to isolate the model's robustness to these perturbations when trained with them. We leave it to future work to extend the forward simulation to incorporate

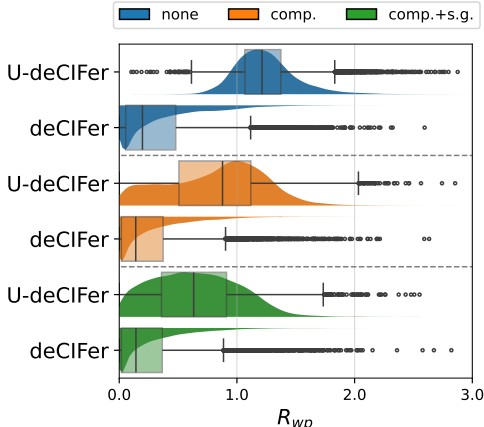

| Desc. | Model | $R_{wp}$ $(\mu \pm \sigma) \downarrow$ | Val. (%) ↑ | MR (%) ↑ |
|---|---|---|---|---|
| **none** | U-deCIFer | 1.24 ±0.26 | 93.49 | 0.00 |
| | deCIFer | 0.32 ±0.34 | 92.66 | 5.01 |
| **comp.** | U-deCIFer | 0.82 ±0.41 | 93.78 | 49.30 |
| | deCIFer | 0.25 ±0.29 | 93.73 | 91.50 |
| **comp.+s.g.** | U-deCIFer | 0.65 ±0.36 | 93.72 | 87.07 |
| | deCIFer | 0.24 ±0.29 | 93.90 | 94.53 |

Figure 3: Left: Distribution of $R_{\mathrm{wp}}$ for deCIFer and U-deCIFer on the NOMA test set with boxplots. Lower $R_{\mathrm{wp}}$ indicates better CIF alignment. Right: Performance for 20K NOMA test samples using deCIFer and U-deCIFer with different descriptors: **none** (no descriptors), **comp.** (composition), and **comp.+ s.g.** (composition + space group). Metrics include validity (Val.) and match rate (MR).

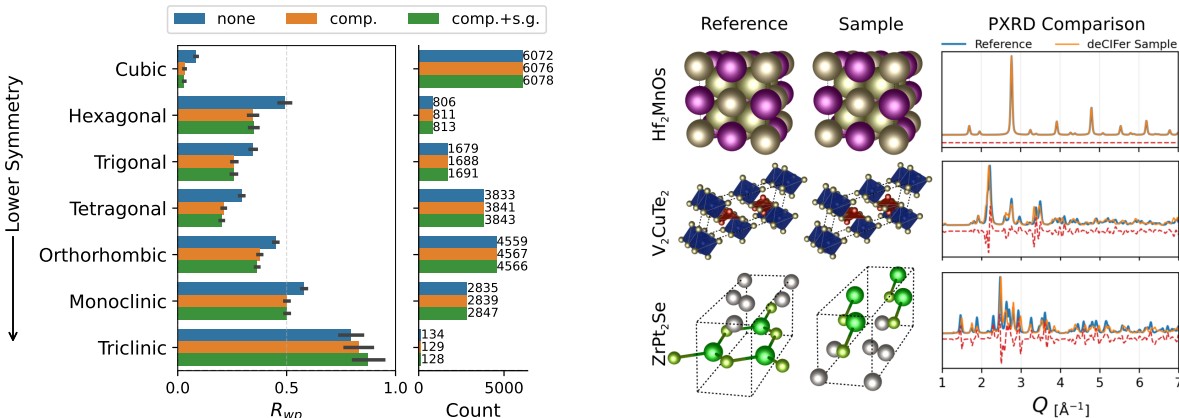

Figure 4: Left: Average $R_{\mathrm{wp}}$ by crystal system for deCIFer on the NOMA test set shows better performance for common high symmetry systems and higher $R_{\mathrm{wp}}$ for rare low symmetry systems. Right: Examples from the NOMA test set highlight this trend with predicted structures from PXRD and composition maintaining reasonable matches even for low symmetry systems with higher $R_{\mathrm{wp}}$.

additional experimental effects, such as background and fluorescence, preferred orientation, absorption and microstrain, peak asymmetry, multi-phase mixtures, and instrument- and preprocessing-specific artifacts.

As summarized in Figure 5, deCIFer remains remarkably stable under ID noise and broadening. Even as the PXRD signal becomes progressively degraded, the model maintains strong alignment with target profiles ($R_{\mathrm{wp}}$) and structure validity. Only under extreme OOD conditions does performance degrade more noticeably. Importantly, this degradation is gradual and consistent with the level of distortion, indicating that the model is not overfitting to narrow PXRD conditions but has learned a robust mapping from signal to structure.

Unsurprisingly, as shown in Appendix Figure A9, lower-symmetry crystal systems remain more difficult to recover under perturbation. But crucially, this behaviour is consistent across conditions, reaffirming that structural ambiguity is a core challenge and not a side effect of noise sensitivity. This supports the view that deCIFer is robust to experimental imperfections at the level of simulation, a critical prerequisite for real-world deployment.

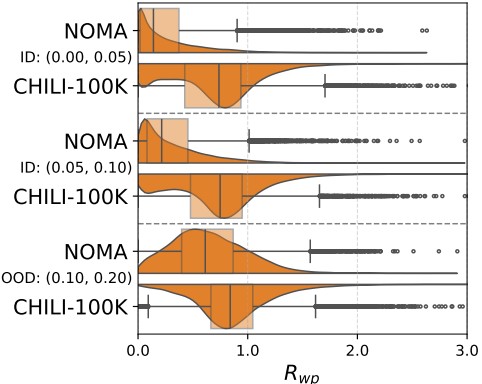

| Dataset (comp.) | Setting | $R_{\mathrm{wp}}$ ($\mu \pm \sigma$) ↓ | Val. (%) ↑ | MR (%) ↑ |
|---|---|---|---|---|
| | U-deCIFer | 0.82±0.41 | 93.78 | 49.30 |
| **NOMA** | ID: (0.00, 0.05) | 0.25±0.29 | 93.73 | 91.50 |
| | ID: (0.05, 0.10) | 0.31±0.30 | 93.77 | 89.28 |
| | OOD: (0.10, 0.20) | 0.65±0.34 | 91.66 | 77.66 |
| | U-deCIFer | 0.96±0.32 | 43.26 | 25.92 |
| **CHILI-100K** | ID: (0.00, 0.05) | 0.70±0.37 | 41.83 | 37.34 |
| | ID: (0.05, 0.10) | 0.73±0.36 | 40.95 | 35.97 |
| | OOD: (0.10, 0.20) | 0.87±0.33 | 33.62 | 26.09 |

Figure 5: Left: Distribution of $R_{\mathrm{wp}}$ for deCIFer on NOMA and CHILI-100K when conditioned on PXRD and composition across three PXRD corruption settings (ID and OOD defined by the noise and FWHM parameters shown). Right: Corresponding summary of $R_{\mathrm{wp}}$, validity (Val.), and match rate (MR). We additionally include U-deCIFer as a PXRD-free baseline for each dataset.

**OOD Evaluation on CHILI-100K:** To test whether this robustness extends beyond synthetic boundaries, we evaluate deCIFer on the CHILI-100K dataset: a curated set of experimentally determined crystal structures with significantly greater structural complexity and lower symmetry than NOMA. CHILI-100K contains no overlap with deCIFer's training data, and thus serves as a structurally out-of-distribution benchmark. Importantly, while the underlying crystal structures in CHILI-100K are experimentally determined, the PXRD patterns used for conditioning are still synthetically generated using the same fixed transformation $\tau_{\mathrm{fixed}}$ defined in Section 3. Consequently, this experiment primarily probes structural out-of-distribution generalization (different chemistry, symmetry, and geometric complexity).

The results are summarized in Figure 5, which also includes U-deCIFer as a PXRD-free baseline for comparison. Despite the increased difficulty, deCIFer maintains a reasonable $R_{\mathrm{wp}}$ and match rate, with only a modest performance drop under in-distribution noise and broadening. As expected, validity decreases mainly due to bond length violations (see Appendix Table A4) but remains interpretable. This highlights the greater geometric complexity of experimental structures. The performance gap relative to NOMA reflects the real structural diversity in CHILI-100K, not failure of conditioning. This is made apparent by comparing this to the unconditioned baseline, where PXRD conditioning leads to a clear improvement in match rate (from 25.9% to 37.3%) and a reduction in $R_{\mathrm{wp}}$, confirming that the gain is not due to memorized compositional priors but driven by alignment with PXRD.

These findings reaffirm that deCIFer's design enables it to generalize beyond synthetic structure datasets and that its failures reflect genuine difficulty rather than collapse. PXRD conditioning proves beneficial not only in idealized simulations but also in settings that approximate real-world structure determination. We reiterate that this evaluation focuses on structural diversity, while real-world PXRD data may contain additional experimental artifacts that could further challenge deCIFer's robustness.

**Consistency and Variability in CIF Generation:** Building on these robustness results, we next examine deCIFer's generative behaviour under repeated sampling from the same PXRD input. Specifically, we investigate the consistency and variability of generated structures when the model is conditioned on a fixed PXRD pattern but with different levels of descriptor constraint. Using a monoclinic structure from the NOMA test set ($Sr_2Cd_2Se_4$), we generate 16,000 CIFs under the three previously established settings: no descriptors ("none"), composition only ("comp."), and composition plus space group ("comp. + s.g.").

Figure 6 illustrates the results. When no crystal descriptors are provided, deCIFer produces a wide variety of cell parameters, compositions, and space groups, reflecting the model's ability to explore the broader space of PXRD-consistent structures. Interestingly, even in this unconstrained mode, the $R_{\mathrm{wp}}$ values are relatively stable, clustering around a narrow range. This suggest that many structurally distinct outputs can yield similar diffraction patterns.

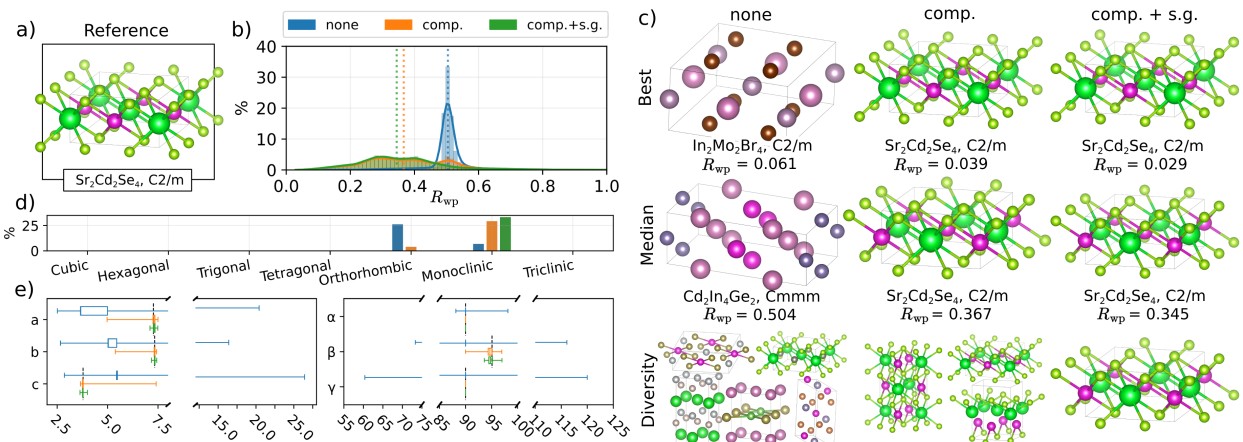

Figure 6: deCIFer-sampled structures for a monoclinic $Sr_2Cd_2Se_4$ PXRD profile (16K samples). a) Reference structure. b) Distribution of $R$wp for generated CIFs. c) Examples of generated structures showing best, median, and diverse samples. d) Distribution of sampled crystal systems. e) Histograms of cell lengths (a, b, c) and angles $(\alpha, \beta, \gamma)$ with reference values as dotted lines.

In contrast, adding composition and space group constraints narrows the distribution of generated structural parameters, as expected, but leads to broader $R_{wp}$ distributions. This highlights the sensitivity of the $R_{wp}$ metric to subtle geometric differences, and highlights the importance of complementing it with other validity or matching criteria. Notably, across all settings, the match rate to the reference structure remains high, indicating that deCIFer can recover accurate solutions both in exploratory and constrained modes.

These findings support a flexible view of deCIFer: when crystal descriptors are known with confidence, adding them improves convergence toward a precise structural solution; when exploring structural hypotheses or navigating ambiguous PXRD signals, descriptor-free generation enables broader sampling of plausible configurations without sacrificing physical fidelity.

# 6 Discussion and Outlook

**PXRD-driven Structure Generation:** The experiments in Section 5 demonstrate that incorporating PXRD as a conditioning signal significantly improves the quality and relevance of generated structures, especially when clear and informative diffraction patterns are available. On the NOMA test set, deCIFer consistently produces structures that align closely with the PXRD target, outperforming unconditioned or composition-only models. This supports a central claim of this work: conditioning on PXRD allows generative models to move beyond compositional priors and directly engage with structural data.

At the same time, these results highlight that PXRD-CSP is inherently more challenging than traditional CSP. When the diffraction signal is ambiguous or when multiple structural solutions (e.g., polymorphs) produce similar PXRD patterns, deCIFer's task becomes fundamentally underdetermined. In such cases deCIFer's performance declines.

Furthermore, our experiments show that deCIFer adapts flexibly to both constrained and unconstrained inference scenarios. When provided with composition or space group descriptors, the model converges to tighter solution distributions. When these are omitted, it samples a broader space of physically plausible structures. This adaptability suggests that PXRD-CSP is not just a harder task, but also a more expressive one, capable of supporting exploratory or targeted workflows depending on available information.

**Extensibility through Conditioning:** By embedding the PXRD signal $\mathbf{y}$ into a learnable conditioning vector $\mathbf{e} = f_{\Phi}(\mathbf{y})$, deCIFer establishes a general and extensible mechanism for incorporating physical measurements into generative modeling. This approach naturally generalizes: if additional data sources are available (e.g., thermodynamic, spectroscopic, or electronic properties), they can be incorporated using sep-

arate conditioning networks. Formally, for $P$ properties $\{\mathbf{y}_1, \ldots, \mathbf{y}_P\}$, the conditional generation objective becomes $\mathcal{L}(\mathbf{X}|\mathbf{Y}_1, \ldots, \mathbf{Y}_P; \boldsymbol{\Theta}, \boldsymbol{\Phi}_1, \ldots, \boldsymbol{\Phi}_P)$. This opens the door to multi-modal structure generation aligned with experimental realities.

**Limitations and Challenges:** While the NOMA and CHILI-100K datasets are stratified and independently curated, data leakage remains a nuanced concern in materials science, where structural or compositional similarity can introduce implicit bias (Cheetham & Seshadri, 2024). However, rigorous preprocessing, deduplication, and independent dataset design significantly reduce this risk.

Another key limitation lies in the nature of PXRD itself. Due to the phenomenon of homometry, where different atomic arrangements produce indistinguishable diffraction patterns, PXRD-informed models cannot always resolve structural degeneracy (Patterson, 1944; Schneider et al., 2010). Metrics like $R_{wp}$ reflect diffraction fit, not atomic uniqueness. Nonetheless, our results show that even partial inclusion of complementary data (e.g., composition) can help disambiguate near-degenerate structures, particularly when combined with fine-grained conditioning mechanisms (Shen et al., 2022).

Finally, while our perturbation and OOD experiments simulate realistic noise and broadening, they do not yet capture the full complexity of experimental PXRD, such as peak asymmetry, background drift, or instrumental artifacts. Addressing these effects will require further refinement of both data simulation and conditioning mechanisms.

**Outlook:** deCIFer and the PXRD-CSP paradigm mark a step toward generative models that do not merely recall statistically likely materials, but actively interpret physical measurements. This makes them fundamentally more aligned with the goals of structure determination in experimental settings. While this approach introduces greater complexity and structural uncertainty, it also makes the generative process more transparent, testable, and useful. Future work can expand on this foundation with richer experimental conditioning, active-learning loops, and downstream applications in materials discovery and verification.

## 7 Conclusion

We introduced deCIFer, a PXRD-conditioned autoregressive language model for crystal structure prediction. Unlike traditional CSP approaches that rely solely on compositional or symmetry priors, deCIFer directly incorporates simulated PXRD profiles as conditioning input, enabling generation of CIFs that are structurally consistent with diffraction measurements. The model is trained on large-scale synthetic datasets and developed with lab-scale compute resources, yet demonstrates robust performance across varying noise levels, peak broadening, and structural complexity, including out-of-distribution generalization.

deCIFer represents a foundational step toward PXRD-informed CSP: a formulation of structure prediction that embraces physical constraints and explicitly addresses the ambiguity inherent in real-world data. While this makes the problem harder, it also makes the model's outputs more interpretable and testable. Our results show that diffraction-guided conditioning substantially improves alignment with structural targets, even when uncertainty or degeneracy is present.

In our comparison to existing state-of-the-art CSP models, we observed that composition-based methods achieve high match rates on benchmark datasets; in large by relying on learned priors that align with frequent structures in the training data. While these models perform well in terms of matching reference structures, they operate under a simpler formulation that does not incorporate experimental constraints. The comparison is therefore currently limited.

In practical terms, deCIFer is best viewed not as an end-to-end solution, but as a powerful structural hypothesis generator. It is particularly effective when the PXRD signal is informative and we suspect that it will be most effectively used in tandem with expert evaluation or downstream refinement. By shifting generative modelling closer to experimental reality, deCIFer lays the groundwork for more integrated and data-aware materials discovery pipelines.

**Acknowledgments:** The authors thank Richard Michael and Adam F. Sapnik for useful feedback.

This work is part of a project that has received funding from the European Research Council (ERC) under the European Union's Horizon 2020 Research and Innovation Programme (grant agreement No. 804066). We are grateful for funding from University of Copenhagen through the Data+ program. RM acknowledges funding provided by the Wallenberg AI, Autonomous Systems, and Software Program (WASP), supported by the Knut and Alice Wallenberg Foundation.

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

# A  Appendix

## A.1  Code and Data Availability

The code for training and using the deCIFer model is open source and released under the MIT License. Official source code and model checkpoints are available here: https://github.com/FrederikLizakJohansen/deCIFer.

## A.2  CIF Syntax Standardization

To enhance the transformer model to process CIFs effectively, we standardized all CIFs in the dataset. Inspired by CrystaLLM (Antunes et al., 2024), we employed similar pre-processing and tokenization strategies, incorporating additional steps to ensure that CHILI-100K (Friis-Jensen et al., 2024) was aligned to the standardized format of NOMA, by the removal certain details such as oxidation states and partial occupancies. We employ the following steps:

1. **Uniform Structure Conversion**: CIFs were converted to `pymatgen.Structure` (Ong et al., 2013) objects to provide a consistent base representation.

2. **Standardized CIF Regeneration**: Using `pymatgen.CifWriter` (Ong et al., 2013), CIFs were regenerated to ensure uniform formatting, eliminate customs headers, etc.

3. **Data Tag Normalization**: The reduced formula, following the `data_` tag was replaced with the full cell composition, sorted by atomic number for consistency.

4. **Symmetry Operator Removal**: Symmetry operators were excluded during pre-processing to simplify the data, but reintroduced during evaluation for validating structural matches. This can easily be done because the reintroduction process uses the space group information retained in the pre-processed files, ensuring compatibility and accuracy.

5. **Incorporation of Extra Information**: Custom properties that are easily derived from the composition of each CIF, such as electronegativity, atomic radius, and covalent radius, were appended to each CIF to maximize the readily available information within each CIF.

6. **Oxidation State and Occupancy Filtering**: Oxidation state refers to the charge of an atom within a compound, which can vary depending on chemical bonding. Occupancy indicates the fraction of a particular atomic site that is occupied in the crystal structure (e.g., a value of 1.0 represents a fully occupied site, while 0.5 indicates partial occupancy). All traces of oxidation states were removed, and only crystal structures with full occupancy were retained. This ensures consistency by aligning CHILI-100K (Friis-Jensen et al., 2024) with the standardized format of NOMA (Antunes et al., 2024).

7. **Numerical Value Normalization**: Numerical values were rounded to four decimal places.

Figure A1 shows a pre-processed and standardized CIF from the NOMA dataset alongside its corresponding unit cell representation and a realisation of its corresponding PXRD profile, as could be input into deCIFer.

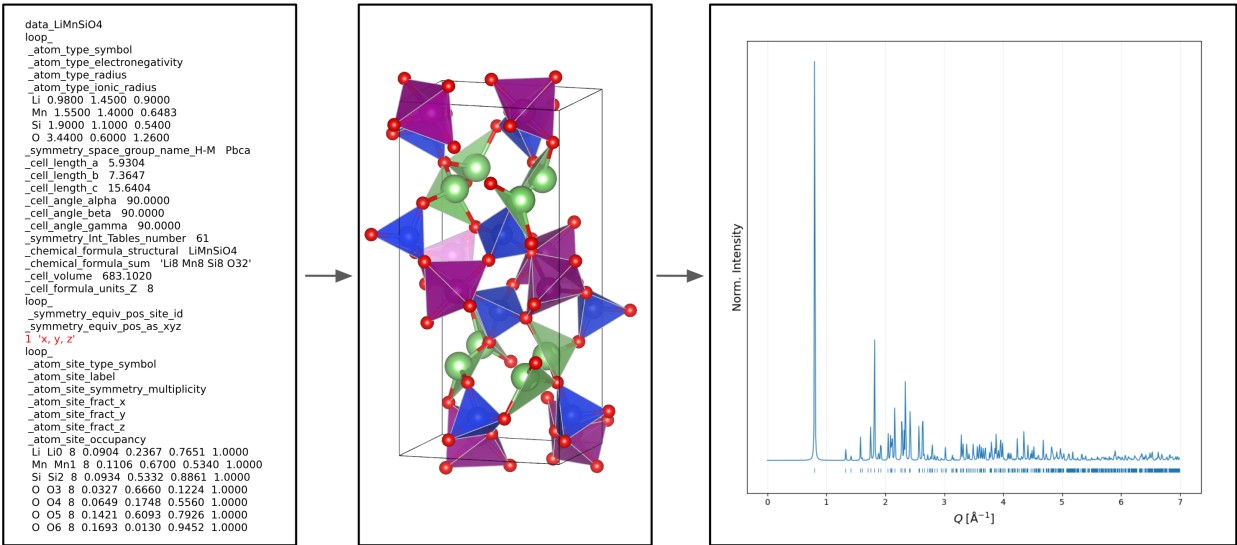

Figure A1: Illustration of a CIF after applying the pre-processing and standardization steps described. Also shown are the corresponding unit cell representation using VESTA (Momma & Izumi, 2008) for visualization and the simulated PXRD profile (with $\sigma^2 = 0.00$ and FWHM=0.01). The red highlight in the CIF indicates where the original symmetry operators were replaced during pre-processing and would be restored for evaluation.

## A.3 CIF Tokenization

To process CIF files effectively, we tokenized each file into a sequence of tokens using a custom vocabulary tailored to crystallographic data in the CIF format. Each CIF was parsed to extract key structural and chemical information, such as lattice parameters, atomic positions, and space group symbols. Numerical values were tokenized digit-by-digit, including decimal points and special characters as separate tokens. Table A1 shows all supported tokens.

Table A1: Supported atoms, CIF tags, space groups, numbers, and special tokens.

| Category | Num. | Tokens |
|---|---|---|
| Atoms | 89 | Si C Pb I Br Cl Eu O Fe Sb In S N U Mn Lu Se Tl Hf Ir Ca Ta Cr K Pm Mg Zn Cu Sn Ti B W P H Pd As Co Np Tc Hg Pu Al Tm Tb Ho Nb Ge Zr Cd V Sr Ni Rh Th Na Ru La Re Y Er Ce Pt Ga Li Cs F Ba Te Mo Gd Pr Bi Sc Ag Rb Dy Yb Nd Au Os Pa Sm Be Ac Xe Kr He Ne Ar |
| CIF Tags | 31 | data__
loop__
_symmetry_space_group_name_H-M
_symmetry_Int_Tables_number
_cell_length_a
_cell_length_b
_cell_length_c
_cell_angle_alpha
_cell_angle_beta
_cell_angle_gamma
_cell_volume
_atom_site_fract_x
_atom_site_fract_y
_atom_site_fract_z
_atom_site_occupancy
_symmetry_equiv_pos_as_xyz
_chemical_formula_structural
_cell_formula_units_Z
_chemical_name_systematic
_chemical_formula_sum
_atom_site_symmetry_multiplicity
_atom_site_attached_hydrogens
_atom_site_label
_atom_site_type_symbol
_atom_site_B_iso_or_equiv
_symmetry_equiv_pos_site_id
_atom_type_symbol
_atom_type_electronegativity
_atom_type_radius
_atom_type_ionic_radius
_atom_type_oxidation_number |
| Space Groups | 230 | P6/mmm Imma P4_32_12 P4_2/mnm Fd-3m P3m1 P-3 P4mm P4_332 P4/nnc P2_12_12 Pnn2 Pbcn P4_2/n Cm R3m Cmce Aea2 P-42_1m P-42m P2_13 R-3 Fm-3 Cmm2 Pn-3n P6/mcc P-6m2 P3_2 P-3m1 P3_212 I23 P-62m P4_2nm Pma2 Pmma I-42m P-31c Pa-3 Pmmn Pmmm P4_2/ncm I4/mcm I-4m2 P3_1 Pcc2 Cmcm I222 Fddd P312 Cccm P6_1 F-43c P6_322 Pm-3 P3_121 P6_4 Ia-3d Pm-3m P2_1/c C222_1 Pc P4/n Pba2 Ama2 Pbcm P31m Pcca P222 P-43n Pccm P6_422 F23 P42_12 C222 Pnnn P6_3cm P4_12_12 P6/m Fmm2 I4_1/a P4/mbm Pmn2_1 P4_2bc P4_22_12 I-43d I4/m P4bm Fdd2 P3 P6_122 Pnc2 P4_2/mcm P4_122 Cmc2_1 P-6c2 R32 P4_1 P4_232 Pnna P422 Pban Cc I4_122 P6_3/m P6_3mc I4_1/amd P4_2 P4/nmm Pmna P4/m Fm-3m P4/mmm Imm2 P4/ncc P-62c Ima2 P6_5 P2/c P4/nbm Ibam P6_522 P6_3/mmc I4/mmm Fmmm P2/m P-4b2 I-4 C2/m P4_2/mmc P4 Fd-3c P4_3 P2_1/m I-43m P-42c F4_132 Pm Pccn P-4n2 P4_132 P23 I4cm R3c Amm2 Immm Iba2 I4 Fd-3 P1 Pbam P4_2/nbc Im-3 P4_2/nnm Pmc2_1 P-31m R-3m Ia-3 P622 F222 P2 P-1 Pmm2 P-4 Aem2 P6_222 P-3c1 P4_322 I422 Pnma P6_3 P3c1 Pn-3 P4nc P-6 P4/mcc I2_12_12_1 P4_2/mbc P31c Ccc2 P4_2/nmc P6_3/mcm C2 Pbca P-4c2 I4_1cd P2_1 P3_112 P4_2mc Pn-3m C2/c R3 P-43m I432 P222_1 I-42d I-4c2 P6cc P6_2 P3_221 P321 Pca2_1 I4_1/acd I4_132 F432 Pna2_1 Ccce Ibca P4/mnc I4_1md P2_12_12_1 R-3c I2_13 P-4m2 Pm-3n I4mm F-43m Pnnm P-42_1c Cmmm P6mm P4_2cm P4_2/m Im-3m Fm-3c I4_1 P4cc Cmme |
| Numbers | 10 | 1 2 3 4 5 6 7 8 9 0 |
| Special | 13 | x y z . ( ) ' , ⟨space⟩ ⟨newline⟩ ⟨unk⟩ ⟨pad⟩ ⟨cond⟩ |

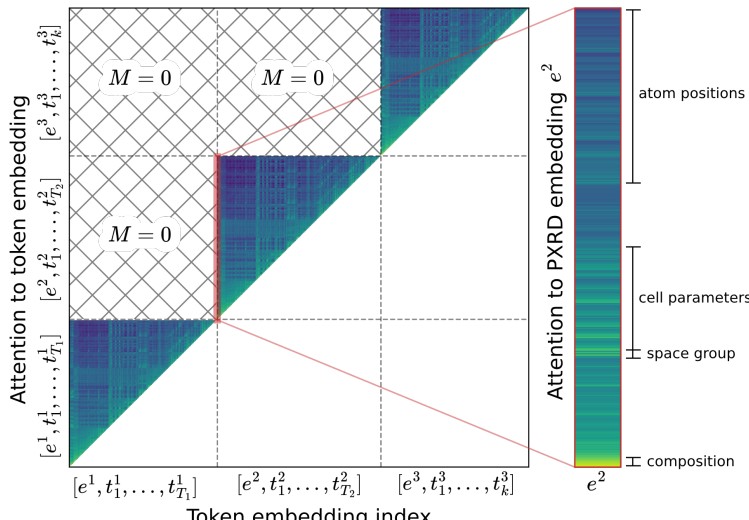

Figure A2: Visualization of the attention masking strategy, showing the log-mean attention weights (averaged over all heads) for an example sequence and highlighting how CIFs are isolated using the attention mask. The figure also illustrates how the embeddings of the second CIF attend to the conditioning PXRD embedding. Lighter shades indicate stronger attention.

## A.4 Attention Masking Strategy

Figure A2 provides a detailed visualization of the attention masking strategy employed in our model. It illustrates the log-mean attention weights (averaged over all heads) for a sample sequence, highlighting the isolation of CIFs through attention masking. The figure also demonstrates how the embeddings of the second CIF attend to the conditioning PXRD embedding. Lighter shades in the figure correspond to stronger attention values.

## A.5 PXRD Simulation

**What do the axes in PXRD mean?** In a typical PXRD experiment, the **x-axis** corresponds to the magnitude of the scattering vector, commonly denoted by $Q$ (in units of Å$^{-1}$), or sometimes the diffraction angle $2\theta$. In this work, we use $Q = \frac{4\pi \sin \theta}{\lambda}$ where $\lambda$ is the radiation wavelength and $\theta$ is the scattering angle. The **y-axis** represents the scattered intensity observed at each $Q$-value, sometimes normalized to have a maximum intensity of 1.

**Peak data and transformations.** Following the methods section, we start with the discrete diffraction peak data: $\mathcal{P} = (q_k, i_k)_{k=1}^n$, where each $q_k$ is the center of a reflection peak, and $i_k$ is the associated peak intensity. To simulate experimental effects, we apply transformation $\tau \sim \mathcal{T}$, which includes **peak broadening** and **additive noise**.

**Peak broadening.** For each peak $k$, centered at $q_k$, we convolve an idealized delta function peak with a pseudo-Voigt profile. At the continuous variable $Q$, the psuedo-Voigt profile is the mixture of a Lorentzian $L$ and a Gaussian $G$, such that

$$\mathrm{PV}_k(Q - q_k) = \eta L(Q - q_k) + (1 - \eta)G(Q - q_k), \tag{2}$$

where $0 \leq \eta \leq 1$ is fixed at $\eta = 0.5$ in this work.

Let FWHM denote the full width at half maximum. The Lorentzian half-width is then $\gamma = \frac{\mathrm{FWHM}}{2}$, making

$$L(Q - q_k) = \frac{1}{1 + \left(\frac{Q - q_k}{\gamma}\right)}. \tag{3}$$

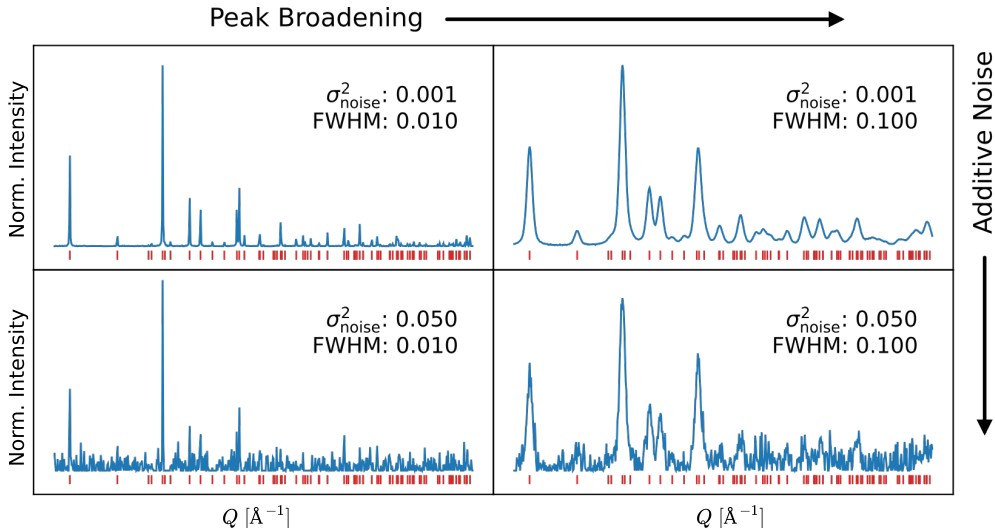

Figure A3: Simulated PXRD profiles with fixed transformation of FWHM and $\sigma^2_{\text{noise}}$ as indicated. Discrete peaks, $\mathcal{P} = \{(q_k, i_k)\}_{k=1}^n$, are shown in red, while the convolved PXRD profiles, $\mathbf{y}$, are shown in blue. Examples with minimal and maximal noise and broadening levels are shown for a compound with composition $CdRhBr_2$ and space group R3m.

The Gaussian standard deviation is $\sigma = \frac{\text{FWHM}}{2\sqrt{2\ln 2}}$, making

$$G(Q - q_k) = \exp\left(-\tfrac{1}{2}\left(\tfrac{Q-q_k}{\sigma}\right)^2\right). \tag{4}$$

**Convolved PXRD.** Given the peak centers $q_k$, itensities $i_k$, and a choice of FWHM, we obtain the convolved PXRD profile

$$I_{\text{conv}}(Q) = \sum_{k=1}^n i_k \, \text{PV}_k(Q - q_k). \tag{5}$$

Afterwards, we normalize $I_{\text{conv}}(Q)$ so that its maximum intensity is 1.

**Noise addition.** Let $\epsilon(Q)$ be drawn from a zero-mean Gaussian distribution with variance $\sigma^2_{\text{noise}}$. This yields the final transformed intensity PXRD profile:

$$I(Q) = I_{\text{conv}}(Q) + \epsilon(Q). \tag{6}$$

**Implementation details.** In practice, we use the `XRDCalculator` from the `pymatgen` library (Ong et al., 2013) for generating the initial discrete peak data $\mathcal{P}$. For training, we sample $Q$-values in $[Q_{\text{min}}, Q_{\text{max}}]$ at increments of $Q_{\text{step}}$. We then apply random transformations $\tau$ during model training. Specific parameters for FWHM and $\sigma_{\text{noise}}$ are listed in Table A2.

### A.6 Validity Metrics

To evaluate consistency and chemical sensibility of the generated CIFs, we conduct a series of validation checks. The methodology is described below.

**Formula Consistency**

We check for consistency in the chemical formula printed in different locations within the CIF. Specifically, we ensure that:

Table A2: Training configuration for deCIFer and U-deCIFer.

| PXRD Transformation Training Parameters | Value |
| --- | ---: |
| Wavelength ($\lambda$) | Cu-K$\alpha$ (1.5406 Å) |
| Q-grid ($Q_{\min}$, $Q_{\max}$, $Q_{\text{step}}$) | (0.0, 10.0, 0.01) |
| FWHM | $\mathcal{U} \sim (0.001, 0.10)$ |
| Mixing Factor ($\eta$) | 0.5 |
| Noise Magnitude | $\mathcal{U} \sim (0.001, 0.05)$ |

| Model / Training Parameters | Value |
| --- | ---: |
| Optimizer | AdamW |
| Learning Rate | $1 \times 10^{-3}$ |
| Warmup Steps | 100 |
| Decay Steps | 50,000 |
| Minimum Learning Rate | $1 \times 10^{-6}$ |
| Weight Decay | 0.1 |
| Batch Size | 32 |
| Gradient Accumulation Steps | 40 |
| Maximum Iterations | 50,000 |
| Embedding Dimension ($n_{\text{embd}}$) | 512 |
| Layers ($n_{\text{layer}}$) | 8 |
| Attention Heads ($n_{\text{head}}$) | 8 |
| Conditioning Model Layers ($n_{\text{c-layers}}$) | 2 |
| Conditioning Model Hidden Size | 512 |
| Sequence Length (block_size) | 3076 |
| Precision | float16 |
| Dropout | 0.0 |

- The chemical formula in the `_chemical_formula_sum` tag matches the reduced chemical formula derived from the atomic sites.

- The chemical formula in the `_chemical_formula_structural` tag is consistent with the composition derived from the CIF file.

**Site Multiplicity Consistency**

We validate that the total multiplicity of atomic sites is consistent with the stoichiometry derived from the composition. Specifically, we ensure:

- The atom types are specified under the `_atom_site_type_symbol` tag.

- The multiplicity of each atom is provided in the `_atom_site_symmetry_multiplicity` tag.

- The total number of atoms derived from these tags matches the stoichiometry derived from the `_chemical_formula_sum` tag.

**Bond Length Reasonability**

To check the reasonableness of bond lengths:

- We use a Voronoi-based nearest-neighbour algorithm implemented in the `CrystalNN` module of `pymatgen` (Ong et al., 2013) to identify bonded atoms.

- For each bond, the expected bond length is calculated based on the atomic radii and the electronegativity difference between the bonded atoms:

- If the electronegativity difference is greater than or equal to 1.7, the bond is treated as ionic, and the bond length is based on the cationic and anionic radii.
- Otherwise, the bond is treated as covalent, and the bond length is based on the atomic radii.

- A bond length reasonableness score $B$ is computed as the fraction of bonds whose lengths are within $\pm 30\%$ of the expected lengths.

- A structure passes this test if $B \geq c_{\text{bond}}$, where $c_{\text{bond}} = 1.0$.

**Space Group Consistency**

We validate the space group by:

- Extracting the stated space group from the `_symmetry_space_group_name_H-M` tag.

- Analyzing the space group symmetry using the `SpacegroupAnalyzer` class in `pymatgen` (Ong et al., 2013), which employs the `spglib` (Togo & Tanaka, 2018) library.

- Comparing the stated space group with the one determined by the symmetry analysis.

**Overall Validity**

A CIF file is deemed valid if all the above checks are satisfied:

- Formula consistency (FM).

- Site multiplicity consistency (SM).

- Bond length reasonableness $B \geq c_{\text{bond}}$, where $c_{\text{bond}} = 1.0$ (BL).

- Space group consistency (SG).

### A.7 Match Rate

The Match Rate (MR) quantifies how many generated structures successfully match their corresponding reference structures, as determined by `StructureMatcher` from the `pymatgen` library (Ong et al., 2013). Two structures are considered a match if their compositions, lattice parameters, atomic coordinates, and symmetry are sufficiently similar, according to the tolerances set in `StructureMatcher`. For the implementation of deCIFer, we follow the example set by CrystaLLM (Antunes et al., 2024), using the parameter values:

- `stol` $= 0.5$: site tolerance, defined as a fraction of the average free length per atom.

- `angle_tol` $= 10°$: maximum angular deviation tolerance.

- `ltol` $= 0.3$: fractional length tolerance, meaning the lattice parameters can differ by up to 30% relative to the reference lattice.

`StructureMatcher` compares two structures by:

- Optionally reducing them to primitive (Niggli) cells.

- Verifying that the lattice parameters are within the fractional length tolerance (`ltol`).

- Checking that the angles are within the angle tolerance (`angle_tol`).

- Ensuring that atomic coordinates align within the site tolerance (`stol`), normalized by the average free length per atom.

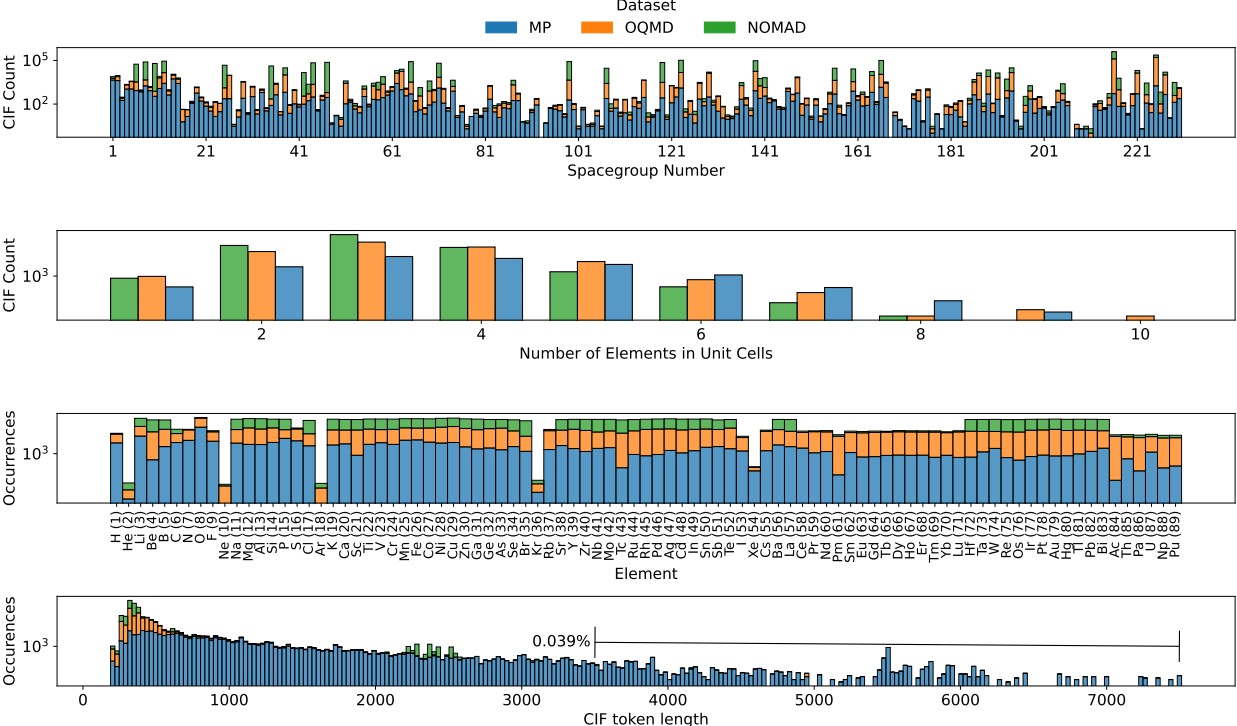

Figure A4: Statistical overview of the NOMA (Antunes et al., 2024) dataset (2,283,346 total samples), showing the distribution of space group frequencies, the number of elements per unit cell, elemental occurrences and CIF token lengths (indicating the percentage of CIFs with larger token sequences than the context length of 3076)

With these parameters, each generated CIF is compared against its reference CIF*. If the two structures are deemed structurally equivalent, we count that as a successful match. MR is computed as the fraction of structures in the dataset for which a match is found:

$$\text{MR} \;=\; \frac{1}{N} \sum_{i=1}^{N} \mathbf{1}\big(\text{match}(\text{CIF}, \text{CIF}^*)\big), \tag{7}$$

where $N$ is the total number of structures and $\mathbf{1}(\cdot)$ is an indicator function that returns 1 if two structures match (according to `StructureMatcher`) and 0 otherwise.

## A.8 Datasets Statistics

Figure A4 illustrates the NOMA dataset. Figure A5 illustrates the statistics of the curated CHILI-100K (Friis-Jensen et al., 2024) dataset.

## A.9 Model Architecture- and Training Details

Table A2 provides a concise overview of all hyperparameters and data augmentation settings used for training deCIFer (and its variant U-deCIFer). Below, we describe additional implementation details.

**Data-stratification** We extract the *space group* number from each CIF (ranging from 1 to 230) and group these into bins of size ten (e.g., 1–10, 11–20, etc.). This heuristic aims to preserve the overall symmetry distribution across splits while reducing the risk of data leakage from structurally similar entries appearing in

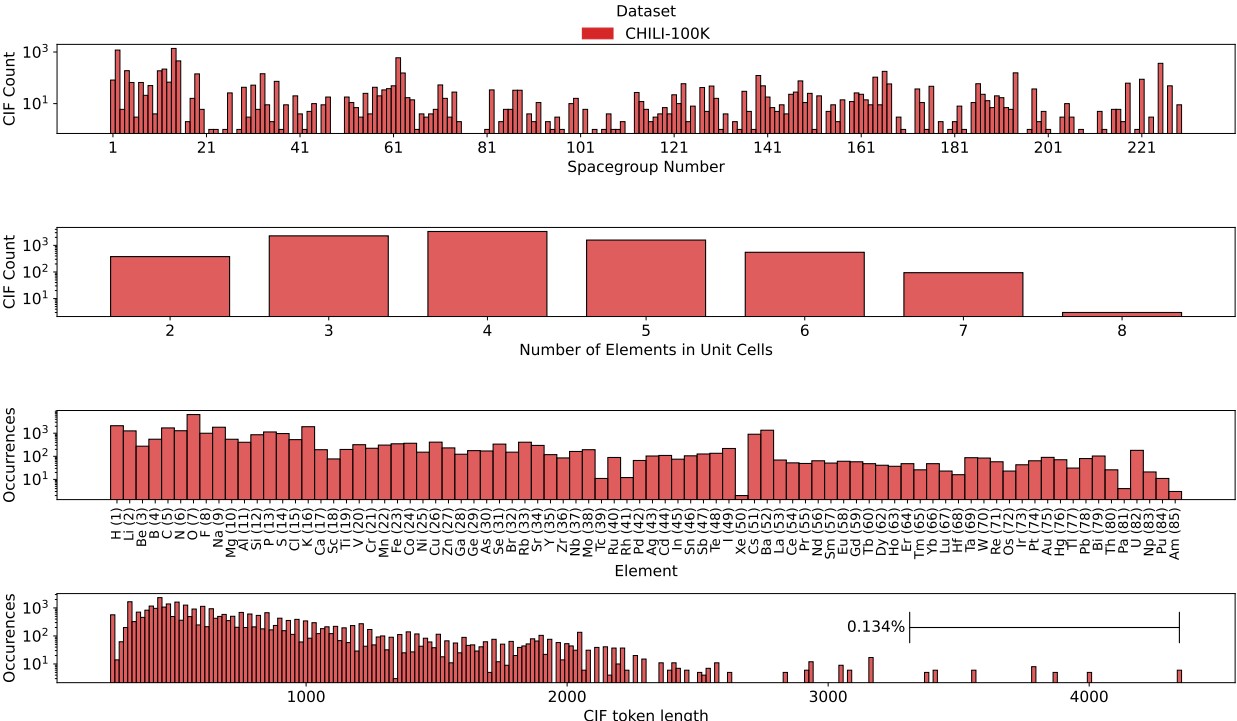

Figure A5: Statistical overview of the curated CHILI-100K ([Friis-Jensen et al.](#), [2024](#)) dataset (8201 total samples), showing the distribution of space group frequencies, the number of elements per unit cell, elemental occurrences, and CIF token lengths (indicating the percentage of CIFs with larger token sequences than the context length of 3076).

multiple subsets. While this does ensures coverage across symmetry classes and even representation of crystal systems across the splits, it does not reflect the most intuitive or principled grouping scheme. In particular, it does not account for finer-grained biases that may be embedded in crystal symmetry or composition distributions. This points to a broader issue in materials datasets: statistical artefacts, such as the "Rule of Four" or symmetry clustering ([Gazzarrini et al.](#), [2024](#)), can introduce shortcuts that models may learn, reducing generalisation and interpretability ([Palgrave](#), [2024](#)). Future work should explore alternative stratification strategies (e.g., stratified sampling based on structural descriptors) to better assess generalisation.

**Hardware Setup** All experiments were conducted on GPUs with sufficient memory to accommodate a batch size of 32 tokenized sequences, each truncated or padded to a context length of 3076. We employed half-precision (float16) to reduce memory usage and improve throughput, ensuring that gradient updates remain numerically stable via built-in automatic mixed-precision.

**Optimizer and Learning Rate Schedule.** We adopt AdamW with a base learning rate of $1 \times 10^{-3}$, which is warmed up for 100 steps and then gradually decayed to $1 \times 10^{-6}$ over 50,000 steps (Table A2). Weight decay is set to 0.1 to regularize model parameters, and we employ gradient accumulation (40 steps) to effectively increase the total number of tokens processed per update.

**Transformer Architectural Notes.** The final transformer stack has 8 layers, each with 8 attention heads, and a model dimension of 512 (embedding dimension). The feed-forward blocks inside each layer use a dimension of $4 \times 512$, and dropout is set to 0.0 to minimize underfitting. We continue to observe stable convergence in practice despite using no dropout.

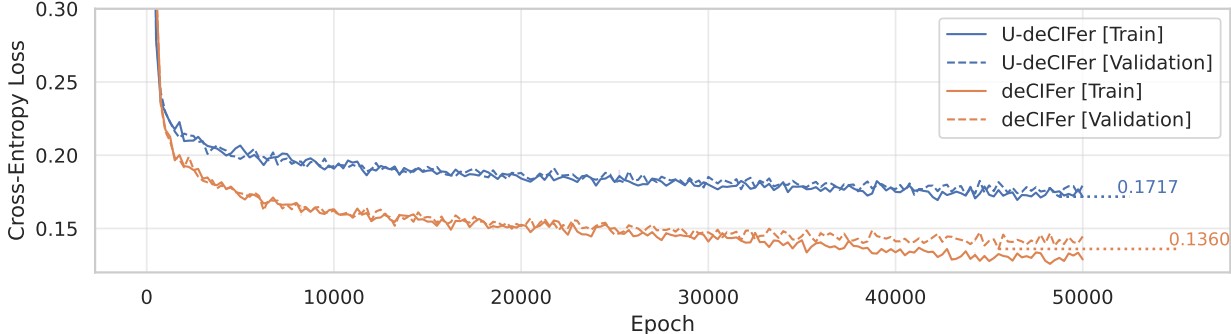

Figure A6: Cross-entropy loss curves for U-deCIFer and deCIFer over 50,000 training iterations, showing progressive reduction in the training and validation losses.

**Maximum Iterations and Convergence.**  We train for 50,000 iterations, at which point the model's cross-entropy loss stabilizes, as illustrated in Figure A6.  Beyond this range, no significant improvements were observed on validation metrics.

### A.10   PXRD Embedding Space

For completeness, we examined the learned embeddings for 50K random training-set PXRD profiles and applied principle component analysis (PCA) for visualization.  As shown in Figure A7, the embeddings form distinct gradients when colored by crystal system, cell-volumes, and constituent atomic numbers $Z$, indicating that the model's PXRD embedding captures relevant structural characteristics, such as symmetry, scale, and elemental composition. These patterns highlight the effectiveness of the conditioning mechanism in encoding meaningful structural information directly from the PXRD input.

### A.11   Baseline Comparison with and without PXRD

With regards to the baseline comparison in Table 1, deCIFer is explicitly conditioned on PXRD data, while the baseline models are conditioned only on composition or latent priors. This distinction means that the comparisons are not direct but instead reveal the relative value of PXRD conditioning. In particular, PXRD conditioning can improve structure prediction when the diffraction signal is rich in structural information, but may introduce ambiguity or conflict with the model's learned priors in cases where the PXRD pattern is noisy or minimally informative. Perov-5 and Carbon-24 provide strong tests of PXRD conditioning due to their polyhedral complexity and carbon-based structural diversity, where diffraction features can directly inform the model. In contrast, MP-20 is drawn from the Materials Project, a dataset where composition-only models may benefit from learned priors due to the high representation of standard chemistries. MPTS-52 further challenges models with low-symmetry structures, where the PXRD signal can become ambiguous, making it difficult for a model to resolve atomic positions purely from the diffraction pattern.

### A.12   Additional Results

Figures A8, A9, and A10 provide additional insights into deCIFer's performance. Table A3 shows a detailsed breakdown of the validity metrics for the NOMA test set corresponding to the results in Figure 3. Table A4 shows a detailed breakdown of validity metrics for the NOMA test set and CHILI-100K test set evaluated on two in-distribution (ID) scenarios and one out-of-distribution (OOD) scenario for the PXRD input.

### A.13   Future Work

One promising area for improvement lies in exploring more advanced decoding strategies, such as beam search, to enhance the generative model's capabilities in downstream tasks. By maintaining multiple hy-

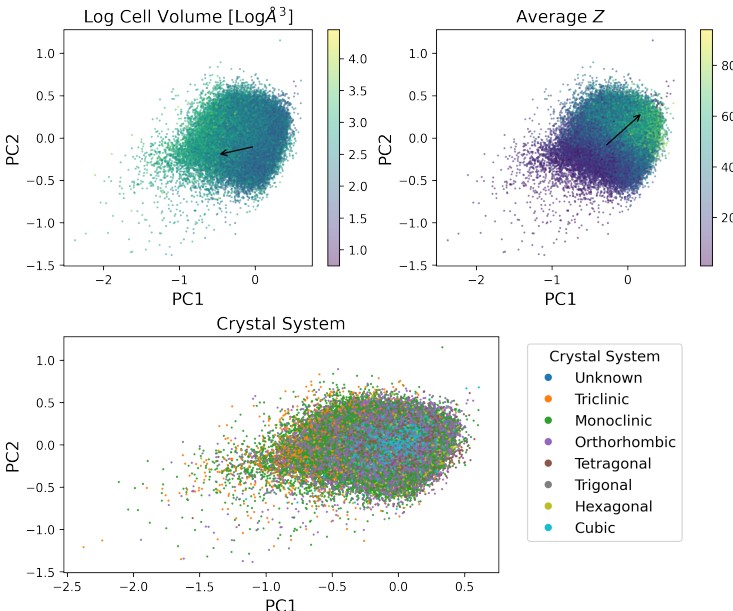

Figure A7: 2D PCA projection of learned PXRD embeddings for 500K training-set samples from NOMA. The three subplots are colored by crystal system, log(cell volume), and average atomic number $Z$, illustrating clear gradients that correspond to structural and compositional features as indicated by the arrows.

potheses during decoding, beam search could produce diverse candidate CIFs for a given PXRD profile, improving structure determination accuracy by ranking outputs based on metrics like $R_{\mathrm{wp}}$. This method could also support optimization strategies that prioritize structural validity and relevance.

Another direction could be to integrate reinforcement learning from human feedback (RLHF) to guide the model more directly toward generating accurate and chemically valid structures (Ziegler et al., 2019). By defining a reward function tailored to properties such as low $R_{\mathrm{wp}}$ values, structural integrity, and adherence to chemical constraints, and interaction with a human expert, RLHF could further refine the model's outputs.

A complementary direction for future improvement lies in increasing the diversity of the training data. While NOMA offers excellent scale, its distribution is skewed toward high-symmetry and well-sampled structures, which limits model generalisation to more uncommon systems. CHILI-100K, with its broader structural complexity and greater representation of low-symmetry crystals, could serve as a valuable training supplement. Mixing synthetic and experimental data, or applying curriculum learning strategies that gradually expose the model to under-represented symmetry groups, could improve generalisation and robustness, particularly for monoclinic and triclinic systems, which remain challenging as seen in Figure 4.

Table A3: Validity of generated CIFs for the NOMA test set using deCIFer and U-deCIFer. Abbreviations: Form = formula validity, SG = space group validity, BL = bond length validity, SM = site multiplicity validity. Overall validity (Val.) is calculated as the percentage of CIFs that satisfy all four validity metrics simultaneously. Match rate (MR) represents the percentage of generated CIFs that replicate the reference CIF.

| **Desc.** | Model | Form (%) ↑ | SG (%) ↑ | BL (%) ↑ | SM (%) ↑ | Val. (%) ↑ | MR (%) ↑ |
|---|---|---|---|---|---|---|---|
| **none** | U-deCIFer | 99.82 | 98.87 | 94.30 | 99.47 | 93.49 | 0.00 |
| | deCIFer | 99.42 | 98.85 | 93.69 | 99.46 | 92.66 | 5.01 |
| **comp.** | U-deCIFer | 99.87 | 99.09 | 94.40 | 99.46 | 93.78 | 49.30 |
| | deCIFer | 99.68 | 99.21 | 94.37 | 99.55 | 93.73 | 91.50 |
| **comp.+s.g.** | U-deCIFer | 99.85 | 98.88 | 94.51 | 99.47 | 93.72 | 87.07 |
| | deCIFer | 99.74 | 99.26 | 94.38 | 99.58 | 93.90 | 94.53 |

Table A4: Validity of generated CIFs for the CHILI-100K test set using deCIFer. Abbreviations: Form = formula validity, SG = space group validity, BL = bond length validity, SM = site multiplicity validity. Overall validity (Val.) is calculated as the percentage of CIFs that satisfy all four validity metrics simultaneously. Match rate (MR) represents the percentage of generated CIFs that replicate the reference CIF.

| **Dataset** | $(\sigma^2_{\text{noise}}, \text{FWHM})$ | FORM (%) ↑ | SG (%) ↑ | BL (%) ↑ | SM (%) ↑ | Val. (%) ↑ | MR (%) ↑ |
|---|---|---|---|---|---|---|---|
| **NOMA** | ID: (0.00, 0.05) | 99.68 | 99.21 | 94.37 | 99.55 | 93.73 | 91.50 |
| | ID: (0.05, 0.10) | 99.64 | 99.18 | 94.39 | 99.55 | 93.77 | 89.28 |
| | OOD: (0.10, 0.20) | 99.60 | 99.87 | 92.60 | 99.49 | 91.66 | 77.66 |
| **CHILI-100K** | ID: (0.00, 0.05) | 95.98 | 97.88 | 42.61 | 94.58 | 41.83 | 37.34 |
| | ID: (0.05, 0.10) | 96.17 | 98.22 | 41.50 | 94.47 | 40.95 | 35.97 |
| | OOD: (0.10, 0.20) | 95.80 | 98.42 | 34.11 | 93.91 | 33.62 | 26.09 |

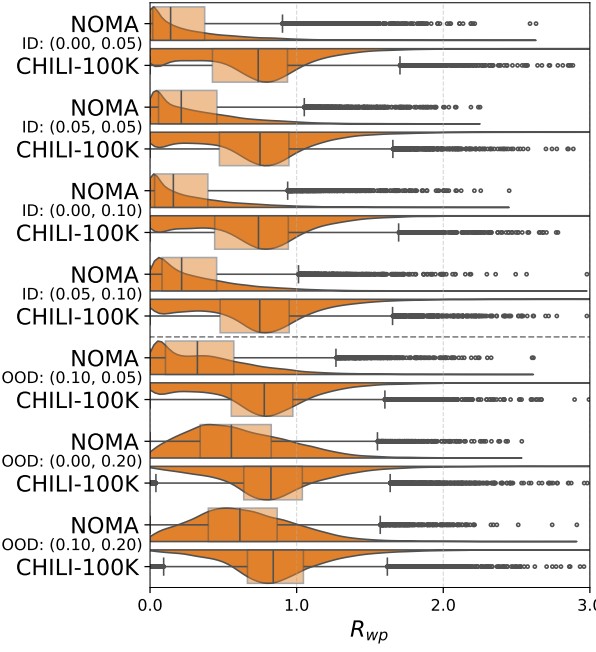

Figure A8: Distribution of $R_{wp}$ for deCIFer on the NOMA- and CHILI-100K test set, presented as violin plots with overlain boxplots; the median is shown for each distribution. Presented are four in-distribution transformations of the input PXRD profiles and three out-of-distribution transformations.

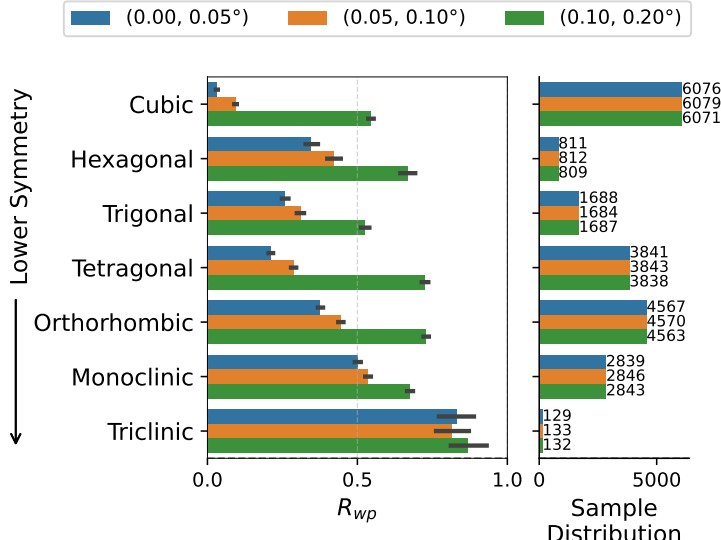

Figure A9: Average metric values by crystal systems for deCIFer on the NOMA test set under two in-distribution transformations of the input PXRD profiles and one out-of-distribution transformation. deCIFer shows better performance for well-represented systems, while rarer, low-symmetry systems lead to worse performance. The right-most plot shows crystal system distribution of the NOMA test set.

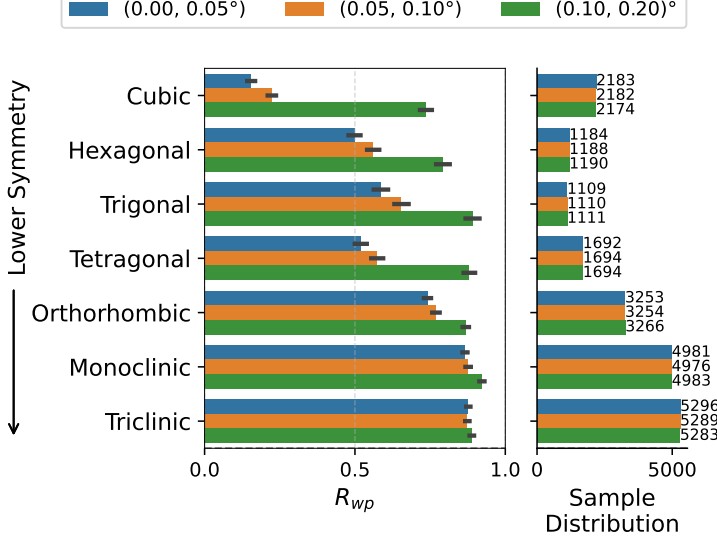

Figure A10: Average metric values by crystal systems for deCIFer on the CHILI-100K test set show better performance for well-represented systems in the training data (NOMA), while low-symmetry systems lead to worse performance. The right-most plot shows crystal system distribution of the CHILI-100K test set, highlighting that CHILI-100K contains a significantly higher proportion of lower-symmetry structures compared to synthetic datasets like NOMA.

