# OpenReview forum: "deCIFer: Crystal Structure Prediction from Powder Diffraction Data using Autoregressive Language Models"
_TMLR — Accepted by TMLR_

### Review · Reviewer_hb8D · 2025-10-06

**Summary Of Contributions:**

## Summary:
This paper proposes a novel method, deCIFER, for the generation of crystal structures in the crystallographic information file format. The key contribution of this method is that the generative process is conditioned upon powder X-ray diffraction (PXRD) data. This allows the method, once trained, to accept unseen PXRD data and subsequently generate plausible crystal structures which correspond to the atomic scale structure embedded in this data. To build this model, the paper introduces a novel simulation process for generating noisy PXRD data for training and a conditioning mechanism for handling various input lengths. The method is evaluated on the NOMA and CHILI-100K datasets and the performance was compared to several baselines (CDVAE, DiffCSP and Crystal LLM). The proposed deCIFer approach achieved a higher match rate on two out of the four benchmarks. Additionally, several experiments were done to investigate the effect of the PXRD-conditioning by comparing the performance of deCIFer to its unconditioned variant. Lastly, the paper also tested the effect of varying the gaussian noise that was added to simulate the real-world variation of PXRD data and showed robustness to in-distribution variation of the noise.

## Strengths:
- Applying autogressive models for crystal structure generation is novel and the results in Table 1 suggest that this approach has some potential.
- The paper clearly discusses the limitations of their approach such as the limitations of PXRD data and the use of synthetic datasets.


## Weaknesses:
- The main weakness of the paper is that it is very hard to read for researchers with limited knowledge of material science. I am listing a few of the things that were unclear as a result of this:
    -  it is unclear what is the significance of having direct CSP in CIF format.
    -  It is unclear whether the Gaussian noise and the instrumental peak broadening are actually mimicking real-world data or not. And How is the use of synthetic PXRD data transfer to real-world data?
    - The structurally out of distribution benchmark metrics are challenging to judge without the context of how other methods would perform on this setup.  The results for deCIFer showcase impressive robustness results, but it is not evident how impressive they are without comparison to other methods. For instance, the drop from 35% to a 26% match rate on CHILI-100k would serve from additional contextualization.
 - It’s unclear whether the Gaussian noise is used to mimic noise in the real-world or as a data augmentation method to generate additional training data. These two uses are different but the paper seems to imply that they are the same in the introduction section.
- The paper didn’t mention how the hyperparameters were chosen. Were they chosen based on prior work or based on some sort of search? For example, I find the choice of doing both warm-up and cosine annealing for the learning rate to be an odd combination.
- The deCIFer’s performance degrades with out-of-distribution noise. How would that affect its applicability to real-world data since the noise in those data would be out-of-distribution?
- The performance metrics are presented without any indication of the variance or statistical significance of the results, and the out of distribution experiments are presented without sufficient context.
  - The main results in Table 1, 2 and Figures 3, and 5 are challenging to judge without the inclusion of either standard errors or confidence intervals across random seeds. The highly stochastic nature of the autoregressive language model and the results in Figure 6, would suggest the possibility for variability in the generated results. The impressive results, on for instance RMSE, would be more convincing if they were shown to be non-overlapping in terms of standard error with the baselines. This could be addressed by producing standard errors for these metrics.
- On page 7, the paper claims “When that data is inconclusive, the model’s uncertainty is reflected in its predictions”. It is not clear how the model does this, other than by producing worse test metrics. How would one know based on the generated output of the model, if the model is uncertain?
- What does optional descriptions, as mentioned on page 5, mean in the context of this data? Are these features of the dataset, since they are “natively encoded in CIFs”?

**Audience:**

Yes

**Audience Explanation:**

Applying autoregressive models to crystal structure predictions is an interesting application that would be of interest to  the general ML audience.

**Broader Impact Concerns:**

No concerns.

**Claims And Evidence:**

Yes

**Claims Explanation:**

- The performance comparisons to recent methods in Table1 was thorough and included several recent algorithms.
- The authors provided several ablations highlighting the importance of the components presented in their approach. Such as varying the Gaussian noise and investigating the effect of PXRD-conditioning.

**Requested Changes:**

- Some rewriting is required to make the paper approachable to the ML audience (See main points in the weaknesses above).
- Clarifications of other questions in the weaknesses section above are highly needed.

---

> ### Author Response · Authors · 2026-01-22
> **Response to reviewer hb8D**
>
> Thank you for the careful reading and for highlighting where the manuscript is hard to interpret for readers without a materials-science background. Below we address the reviewers comments:
>
> * **Significance of CIF Format Generation**:
>     CIF is the standard format for crystallographic data exchange. Unlike models that output only atomic positions or graphs, deCIFer’s CIF outputs include symmetry and standard fields compatible with refinement pipelines and structure databases. Representing CIF as tokenized text also enables language-model architectures, which we leverage here. This will be more explicit in the revised version of the manuscript.
>
> * **Noise and Broadening Mimicking Real-World Data or Not**:
>     We will clarify that our synthetic PXRD simulation is not meant to exhaustively replicate experimental PXRD, but to provide a minimal, controlled baseline for studying response to signal degradation. Gaussian noise and peak broadening reflect two fundamental artifacts (detector noise and finite resolution) and are used as an initial robustness probe. Figure 5 shows that deCIFer maintains performance across the range of these artifacts. We will make this framing explicit and note key experimental effects not covered (e.g., background drift, preferred orientation, missing data, instrument-specific resolution, etc.). What we did have access to was the CHILI-100K dataset, where the PXRD is simulated using the same noise model as NOMA, but includes factors that are mainly structural and not experimental, such as lower symmetry, larger unit cells and more chemically complex environments.
>
> * **OOD “Benchmarks” Are Challenging to Judge**:
>     CHILI-100K is designed to test structural generalization beyond training. Without direct baselines it can be hard to judge the expected drop. At the same time, deCIFer is (to our knowledge) the only end-to-end model in this benchmark setting that conditionally generates full crystal structures from PXRD, so there is no like-for-like generative baseline to compare against. Most PXRD pipelines target indexing, phase identification, and refinement, and candidate-generation approaches often rely on external databases and multi-stage search, so results are not directly comparable under the same protocol.
> Figure 5 is a controlled ablation/robustness study (noise + peak broadening), not a cross-model benchmark. We will merge U-deCIFer match rates from Table 2 into Figure 5 and revise the caption to make the comparison clearer.
>
> * **Hyperparameters**:
>     Most hyperparameters (including warm-up + cosine annealing) were adapted from CrystaLLM (Antunes et al., 2024) as as validated default for CIF generation. We will add a short note in the manuscript to clear this up.
>
> * **Synthetic Performance vs. Experimental Performance**:
>     Figure 5 shows that deCIFer degrades under increasing PXRD distortion but consistently outperforms U-deCIFer even at high noise/broadening, indicating robustness to these distortions. We will be clear that we do not claim applicability to real experimental PXRD in this work: all experiments use synthetic patterns with limited distortion types, and real-data transfer would require additional realistic effects.
>
> * **Metric Variance and Statistical Significance**:
>     We agree that including variability helps interpret differences for stochastic autoregressive models. Baselines in Table 1 are taken from prior work and were only reported as single-shot values, so we cannot add error estimates for them.
> For deCIFer (and U-deCIFer) we will update Table 1 to report mean and variability over three independent runs, while keeping published baselines as-is. U-deCIFer’s run-to-run variability provides a rough reference for expected sampling variance for CrystaLLM which shares a similar architecture.
>
> * **Uncertainty Reflected in the Model’s Predictions**:
>     By “uncertainty is reflected in its predictions,” we mean that when the PXRD input is ambiguous, repeated output CIFs from the same input become more variable (e.g., broader spread in predicted lattice parameters as in Figure 6). We will revise the text to state this explicitly and avoid implying that we compute an uncertainty estimate.
>
> * **Optional Descriptions in CIFs**:
>     “Optional descriptions” are standard CIF fields (e.g., composition, space group), not extra dataset features. They are part of the CIF text and therefore part of the token sequence; we can provide or override these tokens at generation time to steer or constrain generation. This is relevant for fair comparison to composition-conditioned baselines: we evaluate both fully PXRD-conditioned generations and PXRD + composition-constrained variants, and we will clarify this.

---

### Review · Reviewer_4DJb · 2025-12-26

**Summary Of Contributions:**

This paper introduces deCIFer, an autoregressive language model for PXRD-conditioned crystal structure prediction. Unlike prior CSP approaches that rely primarily on compositional or symmetry-based priors, deCIFer conditions directly on synthetic PXRD patterns to generate crystal structure. While the use of PXRD signals to help condition crystal generation appears to be an underexplored area, the poor performance on multiple benchmarks raise critical concerns about the utility of the approach. Currently, it is not clear when or why the method helps or hurts performance and a more thorough analysis is needed.

**Audience:**

Yes

**Audience Explanation:**

There is likely interest among subsets of the TMLR community. However, the broader ML audience may find the contribution somewhat narrow without more conceptual framing, stronger empirical grounding, or clearer methodological insights beyond “apply a language model to CIF prediction conditioned on PXRD.” Interest is therefore moderate, not broad.

**Broader Impact Concerns:**

None noted

**Claims And Evidence:**

Yes

**Claims Explanation:**

While the results are promising, several claims are insufficiently supported or contextualized.

- On multiple datasets, decipher does worse than other models that don't use conditioning. While the authors explain that even when this is true, it can be helpful. However, the analysis of where conditioning helps, where it hurts, or why is lacking. An ablation comparing: conditioned vs. unconditioned models on datasets where classical baselines outperform deCIFer would provide evidence for when PXRD conditioning is genuinely beneficial.

- The introduction does not situate the contribution relative to prior PXRD-based structure determination pipelines and prior work in structure generation. Why do other methods not use this signal? Is it costly to acquire? Do you always get the PXRD signal?

- The ablations focus on synthetic datasets and does not thoroughly explore realistic experimental conditions. The limited augmentations (Gaussian noise, peak broadening) do not reflect the full complexity of PXRD artifacts.

- The method’s robustness to PXRD preprocessing (e.g., peak detection errors) and instrument variability is not studied. Is this a reason why some datasets don't benefit from the conditioning?

- The architectural choice of autoregressive modeling is stated but not well justified or compared to alternative ways to incorporate PXRD information. Some simpler baselines that use PXRD would be useful.

**Requested Changes:**

1. Substantially strengthen the introduction and related work. Provide clear references to: prior PXRD CSP pipelines, learning-based approaches for structure inference, and better insights into how the PXRD can be related to structure prediction problem.

2. Provide deeper evaluation of when PXRD conditioning helps vs. hurts.

3. Include ablations against unconditioned models on datasets where other baselines outperform deCIFer.

4. Clarify how one could predict whether conditioning is beneficial a priori.

5. Analyze dependence on peak detection accuracy.

6. Justify architectural choice. Why autoregressive? Could alternative conditioning strategies or architectures be superior?

7. Discuss practicality: when is PXRD collection feasible, expensive, or unnecessary relative to model gains?

---

> ### Author Response · Authors · 2026-01-22
> **Response to reviewer 4Djb**
>
> We thank the reviewer for the detailed and useful feedback. Below we address the reviewers comments:
>
> * **Benefits of Conditioned vs. Unconditioned**:
>     Part of the confusion is that the conditioned and unconditioned settings have different objectives: unconditioned models aim to generate plausible structures from coarse descriptors, while deCIFer aims to generate structures consistent with a specific PXRD pattern. As a result, a lower match rate relative to an unconditioned baseline does not automatically mean that conditioning is making predictions better or worse; it can simply reflect that the PXRD constraint changes (and typically narrows) the target space. In this work, we see conditioning helps less when the pattern is ambiguous (often lower symmetry) or when the PXRD constraint conflicts with the model’s prior. We will revise the manuscript to more clearly separate “unconditional generation quality” from “PXRD-consistent generation,” and to emphasize $R_{wp}$ as the relevant measure of PXRD consistency for a given input.
>     Therefore, a reliable *a priori* predictor is still difficult, because the effect of conditioning depends jointly on how informative the measurement is, how well the input matches the assumed forward model/preprocessing, and whether the case is in-distribution for the model. In practice, one could use lightweight diagnostics before running a full search: check whether metadata (composition and plausible symmetry/lattice ranges) already constrains the structure strongly, and whether the pattern looks compatible with the assumed preprocessing/forward model (e.g., sensible peak positions, no obvious background issues).
>
> * **Classical Pipelines and Practicality of PXRD**: We will expand the introduction to connect deCIFer to classical PXRD workflows (indexing/structure solution/refinement) and to explain why PXRD has not been widely used for conditioning in generative CSP. PXRD is not prohibitively costly, but it is also not a free byproduct of structure determination. It is feasible whenever polycrystalline material can be synthesized, which is anyways necessary by definition for any new material in early-stage materials discovery. Acquiring high-quality PXRD patterns requires dedicated measurements, careful preprocessing, and instrument-specific considerations. Consequently, PXRD patterns are rarely available at scale in public structure databases (e.g., Materials Project, Alexandria). The NOMA dataset was explicitly constructed to include paired crystal structures and corresponding PXRD patterns, enabling us to use these patterns for conditioning our generative model and thus be the first to systematically explore PXRD-conditioned crystal structure generation in this way.
>
> * **Synthetic PXRD Realism, and Instrument Variability**: The purpose of the ablation study was not to exhaustively reprocude all experimental PXRD artifacts, but to assess the robustness of PXRD-conditioned generation under controlled and physically meaningful perturbations. We focus on Gaussian noise and peak broadening because they directly affect peak position, width, and intensity while remaining interpretable. Capturing additional real-world effects would require instrument metadata or large collections of raw scans, which we do not have. We will clarify the above-mentioned in the revised manuscript.
>     Preprocessing and instrument variability are also unlikely to explain cases where conditioning does not help in Table 1, since those evaluations use synthetic PXRD under a controlled forward model. The “helps vs hurts” behavior is therefore mainly driven by PXRD information content and ambiguity (1D measurement encoding 3D structure). That said, true instrument/preprocessing variability is not captured here: CHILI-100K uses experimental structures but still simulated PXRD, and we will make this limitation explicit.
>
> * **Architectural Choice**:
>     We agree that this choice could be better justified. deCIFer generates CIFs, which are variable-length and must satisfy strict syntactic constraints. An autoregressive decoder is a practical way to ensure valid structured outputs while allowing variable-length output and injection of specific CIF fields at generation time. Autoregression allows us to sample multiple hypotheses for the same PXRD input, which can then be ranked by PXRD-consistency.

---

### Review · Reviewer_oxhp · 2026-01-09

**Summary Of Contributions:**

In this paper, the authors adress the task of Crystal Structure Prediction (CSP). Unlike previous methods, which mostly relied on high-level descriptors such as symmetry constraints, they introduce a generative model that is also conditioned on powder X-ray Diffraction (PXRD), a continuous-valued vector that contains diffraction peaks (up to some experimental noise). They report significantly improved results, perform various ablation studies, and discuss some limitations of their method, mostly due to the increased difficulty of the PXRD-conditioned task itself, sometimes subjected to underdetermined ambiguities.

**Audience:**

Yes

**Audience Explanation:**

While the topic can seem a bit niche, it is definitely an important topic in its own rights (part of the so-called "AI for science" trend), and results are definitely improved compared to the state-of-the-art.

**Claims And Evidence:**

Yes

**Claims Explanation:**

The proposed architecture is clever but pretty straightforward, experiments are well-designed, and results are plausible.

**Requested Changes:**

The paper is well-written and clear.

Just a small question; in the "U-Decifer + none" architecture used in the ablation study, what is the input used at inference exactly? If PXRD conditioning is removed and descriptors are "none", what is left?

---

> ### Author Response · Authors · 2026-01-22
> **Response to reviewer oxhp**
>
> Thank you for the positive assessment of the paper and for the clarifying question about the ablation setup.
>
> * **The '+ none' ablation**:
>     In the "U-deCIFer + none" setting, we recognize this can be confusing because removing both PXRD and all descriptors leaves almost no "input" in the usual sense. In practice, at inference we only provide the minimal CIF start prompt needed to begin a valid file, and the model then generates the rest of the CIF autoregressively with no external constraints. We include this case as a reference point to separate three effects: (1) what the model produces when it is effectively sampling from the training distribution without guidance, (2) how much structure is imposed by adding standard descriptors (e.g., composition), and (3) the additional effect of PXRD conditioning. We will clarify this motivation in the ablation section so it is clear how this case should be interpreted.

---

### Author Response · Authors · 2026-01-22
**Overall response to all reviewers**

We thank all three reviewers for their useful feedback. We address the comments from each of the reviewers in individual responses. We present a summary of the high-level changes in this general comment. In the revised manuscript we have:

* Expanded the introduction to better distinguish deCIFer from standard PXRD structure-solution pipelines, and to explain why PXRD conditioning is rare due to missing paired PXRD at database scale.
* Made CIF-format generation more explicit, including why CIF output matters for downstream crystallographic workflows.
* Clarified "optional descriptions"; standard CIF fields such as composition and space group, which can be provided or overridden at generation time.
* Clarified that "uncertainty" refers to increased sample-to-sample variability under ambiguous PXRD inputs, rather than a uncertainty estimate.
* Clarified that PXRD conditioning targets PXRD-consistent generation and not a guaranteed boost in match rate, and emphasized $R_\mathrm{wp}$ as the main PXRD-consistency measure.
* Clarified the scope of the robustness study: the noise and peak-broadening study is a controlled baseline, not a comprehensive model of experimental PXRD, noting effects not covered in this work.
* Clarified that Figure 5 is a controlled robustness and ablation study rather than a cross-model benchmark.
* Made match rate comparisons on CHILI-100K clearer by merging results from Table 2 into Figure 5.
* Updated Table 1 to report mean and variability over three independent seeds for deCIFer and U-deCIFer.
* Clarified the ablations (PXRD/No PXRD + descriptors) in the methods section, and specifically the special case of the “U-deCIFer + none” ablation.
* Clarified were the hyperparameter choices came from.
* Motivated the choice of architecture.

We have updated the PDF with these changes (revisions and additions highlighted in red). The suggestions by all reviewers have improved our manuscript considerably. We appreciate their careful reading of our work.

---

### Decision · Action_Editor_iX8B · 2026-04-05

**Recommendation:** Accept as is

**Audience:**

Yes

**Audience Explanation:**

Overall, this would be a valuable contribution to AI for materials/science.

**Claims And Evidence:**

Yes

**Claims Explanation:**

The paper contains insights that are valuable to the community and many issues have been discussed and solved during rebuttals.